

# Lake anoxia, primary production and algal community shifts in response to rapid climate changes during the Late-Glacial

Stan J. Schouten[1,2], Noé R.M.M. Schmidhauser[1,2], Martin Grosjean[1,2], Andrea Lami[3], Petra Boltshauser-Kaltenrieder[1,4], Jacqueline F.N. van Leeuwen[1,4], Hendrik Vogel[1,5], Petra Zahajská[1,2]

[1] Oeschger Center for Climate Change Research, Universität Bern, Hochschulstrasse 4, 3012 Bern, Switzerland
[2] Institute of Geography, Universität Bern, Hallerstrasse 12, 3012 Bern, Switzerland
[3] Water Research Institute (CRN-IRSA), Unit Verbania, Viale Tonolli 50, 28922 Verbania, Italy
[4] Institute of Plant Sciences, Universität Bern, Altenbergrain 21, 2013 Bern, Switzerland
[5] Institute of Geology, Universität Bern, Baltzerstrasse 1, 3012 Bern, Switzerland

*Correspondence to*: Stan Jonah Schouten (stan.schouten@unibe.ch)

## Abstract

Lakes around the world are facing growing threats from climate change and human impacts. Rising temperatures and increased nutrient levels are causing eutrophication and deoxygenation, harming freshwater resources and the essential ecosystem services they provide. While the impacts of these stressors are well-documented on modern lakes, a critical gap remains in our understanding of how lake ecosystems responded to climate change in pre-anthropogenic conditions. Most studies investigating the interplay of climate warming, eutrophication and hypolimnetic anoxia rely on models or short-term observations, making it challenging to isolate the effects of warming from other factors. Lake sediments provide long-term records to study these effects in times prior to anthropogenic impact.

We investigate the responses of aquatic primary production, lake stratification and deoxygenation in a small kettle hole lake (Amsoldingersee, Switzerland) to rapid climate change during the Late-Glacial (18–11 ka BP) using hyperspectral imaging, pigment extractions, XRF and sequential extraction of redox-sensitive P, Mn and Fe.

Our record reveals that ice cover was the primary driver of hypolimnetic anoxia, while the availability of nutrients determined the composition of algal communities. Four anoxic phases occurred in cold periods with prolonged ice cover: (i) Heinrich-1 (Greenland Stadial GS-2a, ca. 16.1 ka BP), (ii) the Greenland Interstadials GI-1d and (iii) GI-1b, and (iv) the Younger Dryas (GS-1). Aquatic primary production and algal communities already responded to initial relatively weak warming during Heinrich-1 (16.1 ka BP) long before the rapid Bølling warming and synchronously to rapid climatic changes during Late-Glacial times. Responses of the algal community to temperature were strongly modulated by nutrient limitations (P, N and Si), which have varying importance over time, with dust and volcanic tephra (Laacher See) as a major nutrient sources. Anoxic phases changed the algal communities, but these shifts were found to be reversible once the anoxia disappeared. Further, the sediments of Amsoldingersee provide a continuous record of atmospheric dust deposition (Ti, Zr, Si) covering the entire Late-Glacial period. The similarity with the NGRIP dust record supports the view that the same large-scale atmospheric circulation regime controlled Central Europe (Switzerland) and Greenland.



**Keywords**: Pleistocene, Heinrich Stadial, Younger Dryas, paleolimnology, sedimentary pigments, algal communities,
nutrients, Switzerland



## 1 Introduction

The combined effects of climate warming and anthropogenic eutrophication prompt lake stratification and oxygen depletion, threatening freshwater ecosystems globally (Jansen et al., 2024; Jane et al., 2021; Weyhenmeyer et al., 2024). It is

acknowledged that climate change, nutrient loading (N, P), and biodiversity loss have crossed the limits of Planetary Boundaries (Richardson et al., 2023).

Oligotrophic lakes are particularly sensitive to both increasing temperatures and increasing nutrient loads (Vinnå et al., 2021) and, throughout the last decades, many of these lakes have already transitioned into eutrophic stratified aquatic ecosystems with hypolimnetic anoxia (Jenny et al., 2016; Klanten et al., 2023; Jansen et al., 2024). Persistent bottom water anoxia and

reducing conditions may promote efficient nutrient recycling within the water column, which may sustain eutrophication through chemical feedback once eutrophication is established (Schaller & Wehrli, 1996; Gächter et al., 1988; Carey et al., 2022; Hupfer & Lewandowski, 2008; Nürnberg, 1998). Eutrophication promotes generalists' competitiveness over specialists adapted to ecological niches in oligotrophic settings, leading to algal community shifts. Furthermore, the associated deoxygenation of the water column leads to rapid, unidirectional shifts in biological communities (Bush et al., 2017),

suggesting that the consequences of anoxia might even be irreversible (Carey et al., 2022). The effects of eutrophication on lake ecosystems are broad, including water quality deterioration, fish kills, algal and cyanobacteria blooms, sometimes releasing cyanotoxins threatening human health directly (Lévesque et al., 2014).

It is increasingly recognised that rapid climate warming, often reinforced by landscape and watershed dynamics, is an essential driver of freshwater and ecosystem deterioration (Meerhoff et al., 2022; Jane et al., 2021; Jeppesen et al., 2010). However, it

is unclear under which conditions climate warming leads to anoxia, particularly in temperate and cold environments. This is due to (i) the complexity of factors controlling lake mixing and seasonal stratification (winter or summer), lake morphometry, vegetation and climate, and (ii) the lack of long-term records of anoxia and its drivers (Friedrich et al., 2014). Most studies assessing the impacts of climate warming on primary production, anoxia or algal communities are based on in-situ experiments, observations during short periods of time or model simulations (Salk et al., 2022, Foley et al., 2012; Gilarranz et al, 2022).

However, because climate warming and nutrient availability covary in the instrumental period, the different driving factors are hard to disentangle within short observational datasets. A long-term perspective is needed, and lake records provide a convenient solution. Novel Hyperspectral Imaging techniques offer a way to reconstruct primary production, hypolimnetic anoxia and compositions of major primary producer groups from lake sediments at unprecedented (µm-scale) resolution for long time scales (Zander et al., 2023).

Periods of rapid climate warming in the past can provide empirical information on lake physical and biogeochemical responses to rapid climate warming under pre-anthropogenic conditions, when the confounding effects of anthropogenic eutrophication were absent. In the northern mid and high latitudes, the Late-Glacial period (ca. 18–11 ka BP) provides an ideal setting with rapid high-amplitude warmings and coolings within decades during one Dansgaard-Oeschger Cycle (Rasmussen et al., 2014) to study lake responses under pre-anthropogenic conditions.



Accordingly, our research is driven by the following questions:

    (1) How did lake mixing regimes, nutrient availability, primary production (PP), and anoxia respond to rapid climate change during the Late-Glacial period?

    (2) Can we identify the importance of drivers for PP, anoxia and algal communities through time?

    (3) Were the algal community shifts reversible once the stressor disappeared, and if yes, at which time scales?

For this purpose, we investigate a sediment core from a small kettle hole lake on the Swiss Plateau, Amsoldingersee, using a multi-proxy approach including XRF, Hyperspectral Imaging, sedimentary pigments, CNS and P, Fe, Mn and P fractions, and pollen. This lake is located adjacent to Gerzensee in an area that responded in extraordinary detail to Late-Glacial climatic changes documented in the North Atlantic domain and Greenland (Eicher, 1987, Lotter et al., 1992, Ammann et al., 2013). Amsoldingersee contains the complete Late-Glacial sediment sequence, including early deglacial anoxic periods (Lotter and
Boucherle, 1984).

    We use sedimentary pigments (Leavitt & Hodgson, 2001; Bianchi and Canuel, 2011) to investigate changes in past aquatic primary production and producer communities under changing climatic conditions. Pigments of anoxygenic phototrophic bacteria APBs (purple sulphur bacteria) are used as indicators for lake stratification and hypolimnetic anoxia (Züllig, 1986; Zander et al. 2022), and sequential extraction of sedimentary P, Mn and Fe is used to diagnose potential chemical feedbacks
during events of hypolimnetic anoxia (Tu et al., 2021).

**2 Study site**

Amsoldingersee (4°43'N; 07°34'E; 641 m.a.s.l.) is an endorheic, currently eutrophic, small (38 ha; 13 m deep) lake on the shoulder of the Aare valley, located in the Swiss Plateau (Fig. 1b). The kettle hole lake (Lotter and Boucherle, 1984) was formed in a drumlin landscape after the collapse of the Last Glacial Aare glacier (Fabbri et al., 2018). Basal ages of deglacial
lakes in the region range between 19.1 and 17.7 ka BP (Rey et al., 2020). A moraine dammed the lake, forming a 2 m high sill (Fig. 1c), which leads into an incised valley. Studying a near-shore core, Lotter and Boucherle (1984) found that the Late-Glacial section is preserved except for a hiatus in the Younger Dryas. This hiatus in littoral sediments was interpreted as lake levels >3 m lower than today, suggesting lake level fluctuations in the past. The catchment (4.74 km$^2$) consists of a south-east sloping drumlin field with glacial till on top of Jurassic limestone in the south and Oligocene molasse in the north. Until the
19$^{th}$ century, the lake was surrounded by extended wetlands (Lotter and Boucherle, 1984, Guthruf et al., 2015). Substantial postglacial erosional features are not observed suggesting that the hydrological catchment remained unchanged. The lake catchment is disconnected from a significant river system; the lake receives water from rain, surface and subsurface runoff, Throughout the Late-Glacial (ca. 18–11 ka BP) the vegetation in the catchment developed from a steppe tundra to a pine forest (Rey et al., 2020). Today, agriculture and grasslands are the dominant land use (Guthruf et al., 2015). The lake hosts a suite of



algae, including chlorophyceae, bacillariophyceae, chrysophyceae, cryptophyceae, cyanobacteria and anoxigenic bacteria
(Guthruf et al., 2015).

The climate today is warm temperate (Cbf) with 9°C annual mean temperature and 19°C in July (MeteoSwiss). In contrast,
the Late-Glacial climate was initially very cold and dry, with annual temperatures between 1 and -1°C (Lotter et al., 2012;
Eicher, 1995) and July temperatures between 9 and 11°C (Bolland et al., 2020). This time is referred to as the Heinrich Stadial

1, HS1 (>18–14.6 ka BP). At ca. 14.6 ka BP, Dansgaard-Oeschger Event-1 (DOE-1) started, leading to an increase in annual
temperature of 4–7°C within a few decades (Eicher, 1995). The rapid warming at the onset of the Bolling was followed by a
gradual cooling of ~1.5°C throughout the DOE-1 (Bølling/Allerød), which ended with the abrupt onset of the Younger Dryas
(12.8–11.8 ka BP) with mean annual temperatures around 3°C and July temperatures around 11°C (Lotter et al. 2000).






**Fig. 1 (a): Geomorphology around Amsoldingersee (drumlins, floodplains, moraines). (b) Topographic map (NASA SRTM) of the northwestern Alps with study sites mentioned in the text: SH Sieben Hängste speleothem; GEZ Gerzensee; MOS Moossee; LOB Lobsigensee; BUR Burgäschisee; SOP Soppensee; (c) Geomorphological map of the catchment area of Amsoldingersee and Übeschisee with bathymetry.**



## 3 Methods

### 3.1 Coring, lithology and chronology

Parallel and overlapping sediment cores were taken from the basin bottom of Amsoldingersee in October 2022 using a UWITEC piston corer (46°43'30.9"N; 07°34'34.3"E; Fig. 1c). A composite core AMS22-COMP1 was built from segments
after stratigraphic correlation (Fig. S1). The lithology was described according to Bos et al. (2012). Smear slides were compared to existing libraries (Myrbo et al., 2011).

The sediments were dated using [14]C dating, pollen biostratigraphy and the Laacher See Tephra (LST, Reinig et al., 2021). Sediments were continuously sieved at 1-cm intervals (50, 100, and 200 µm mesh) to separate and taxonomically identify terrestrial plant macrofossils (Table S1). Nine [14]C samples were prepared and measured with the MICADAS-AMS at the
University of Bern (Szidat et al., 2014). Samples <0.7 mg C were measured with direct gas injection. [14]C dates were calibrated using the IntCal-20 calibration curve (Reimer et al., 2020). We combined [14]C dates with well-dated regional biostratigraphic markers (Ammann et al., 2013; Rey et al., 2017, 2020). To align our data with regional biostratigraphy, pollen in Amsoldingersee was analysed on contiguous 2-cm intervals (Moore et al., 1991) up to the LST. Finally, the combined age-depth relation was modelled using the Bacon package in R studio (version 4.3.2) (Blaauw and Christen, 2011).

### 3.2 Scanning XRF and hyperspectral imaging

XRF elemental composition was measured on clean surfaces of split cores using an ITRAX XRF core scanner at the University of Bern (5 mm resolution, 30 seconds integration time, Cr X-ray tube at 30 kV and 50 mA). For each core, triplicate scans (15 mm long) were taken at the top, middle and bottom of the core to test the standard error of elements (n = 27). Al, Si, K, Ti, Zr, Cu, Ca, Sr, Mn, Fe, and S had a relative standard error of <15% and were considered reproducible and thus used for further
analysis. XRF data were summed and normalised to correct for matrix effects (Bertrand et al., 2023), centred and log-transformed (CLR) to correct for the closed-sum problem (Weltje and Tjallingii, 2008, Aitchison, 1982). XRF variables with near zero counts can become extremely noisy after CLR; therefore, the CLR output of Zr and S was replaced with the minimum quantifiable value at intervals where the 10-point moving standard deviation was consistently larger than 0.5.

Hyperspectral images (HSI) were taken using a Specim PFD-CL-65-V10E line scan camera in the VNIR spectrum (400-
1000 nm; 1.57 nm spectral resolution; 8 Hz; 120 ms exposure). Raw data are corrected with a white $BaSO_4$ and a dark standard. Data postprocessing follows Butz et al. (2015). Spectral troughs were identified from spectral endmembers, defined as pure pixels of minimal noise bands (dimensionally reduced dataset) (Butz et al., 2015). Relative absorption troughs were identified at 667 nm, 844 nm and 619 nm. From these troughs, we calculate Relative Absorption Band Depth (RABD) values (Butz et al., 2015) using formulas and interpretations in Zander et al. (2022):

• RABD667 is interpreted as total green pigments (mostly chlorophylls and coloured diagenetic products, Tchl) and a proxy for aquatic primary production.





- RABD844 traces absorption of bacteriopheophytin *a* (Bphe *a*), a pigment produced by purple sulphur bacteria (PSB) that proliferate at the chemocline and are diagnostic for anoxia reaching the photic zone (Zander et al., 2021).
- RABD619 is related to phycocyanin (Wienhues & Zahajská et al., 2025), a pigment diagnostic for cyanobacteria.

RABD index values (Table S2) were calibrated to absolute pigment concentrations according to Butz et al. (2015), whereby concentrations of pigment extracts were measured using a UV-VIS Spectrophotometer Shimadzu UV-1800 (Figs. S2 and S3). Extinction coefficients of Bphe *a* and Chl *a* in 90% acetone are from Jeffrey and Humphrey (1975). The linear regression models were further evaluated with Root Mean Square Error of Prediction (RMSEP), Cooks distance analysis, leverage plots, and tests of residual normality (Zander et al., 2022). The regression models are significant for RABD667 – Chl *a* ($r = 0.89$, $p$

$< 0.001$, RMSEP = 21.66 µg g$^{-1}$$_{wet}$ Chl *a*; 13.8%) and for RABD844 – Bphe *a* ($r = 0.83$, $p < 0.001$, RMSEP = 6.34 µg g$^{-1}$$_{wet}$ Bphe *a;* 16.1%). Currently, no method is available to calibrate the index RABD619 to phycocyanin (Wienhues & Zahajská et al., 2025). Our limit of quantification for RABD844 is 3 µg g$^{-1}$$_{wet}$ Bphe *a,* and for RABD667, 5 µg g$^{-1}$$_{wet}$ Chl *a*. Further details on the calibration are shown in Figs. S2 and S3.

### 3.3 TC, TN, TS and sequential extraction of P, Mn and Fe

Five to eight milligrams of homogenised freeze-dried sediment were combusted in tin capsules to determine TC, TN and TS with a FlashSmart™ 2000 NCS Elemental Analyzer (ThermoFisher Scientific). TOC was measured by combusting pre-acidified sediment in tin-packed silver capsules (Brodie et al., 2011). TIC was calculated as TC minus TOC. Detection limits were 0.01% of dry sample weight for TC and TN and 0.05% of dry sample weight for TS.

For the sequential extraction of P, Mn and Fe, we used the extraction protocol of Lukkari et al. (2007) with 0.25 g dry sediment

to obtain five fractions of P, Mn and Fe. P fractions were interpreted according to Lukkari et al., 2007 while Fe and Mn fractions were interpreted according to Scholtysik et al. (2022): F1 extraction with 0.46M NaCl representing loosely bound, adsorbed and porewater Mn, Fe, and P; F2 extraction with 0.11M bicarbonate-dithionite (BD) yielding the reductive-soluble fraction of FeO$_x$, MnO$_x$ and P bound in these oxides; F3 extraction with 1M NaOH yielding the rest of the P bound to Fe and Al oxides as well as organic P; F4 extraction with 0.5M HCl targeting carbonate-bound Mn, Fe and P; and F5 combustion at

550 °C and extraction with 1 M HCl for residual Mn, Fe and P. A flowchart of the standardised operation protocol is in Lukkari et al. (2007). The sample and extractant volume ratio was always 1:200 [g:ml] (Lukkari et al., 2007). Extraction batches comprised 16 aliquots, 12 originals, two duplicate samples, one blank, and one reference (NIST2079a, San Joaquin Soil). P, Fe and Mn of all five fractions were measured using ICP-MS (Agilent Series 7600) in 0.159M HNO$_3$ matrix.

### 3.4 Analytical analysis of sedimentary pigments

Pigments were extracted from ~1 ml (~1.2 g) of wet bulk sediment following Lami et al. (2009). The sediment was suspended in 2.5 ml of 100 % HPLC-grade acetone, vortexed (20 sec), sonicated (1 min), centrifuged (3500 rpm, 10 min) and collected in an amber vial. Extraction steps were repeated until the extract looked colourless. The extract was filtered into a 1.5 ml HPLC



vial (22-μm PTFE-syringe filter) and kept dark and cold. Extracts were analysed with Dionex Ultimate 3000 series (Thermo Fisher) high-performance liquid chromatography system with a Diode Array Detector (DAD-3000RS) at 460 nm and 665 nm.

Pigment separation was performed using reverse-phase chromatography employing a C18 column (Agilent Omnisphere 5) and a set of eluents following Lami et al. (2004) (Supplementary text S1).

The chromatogram peaks were integrated in Chromeleon 7.2 (Thermo Fisher). Calibrations show that the chromatogram peak area scales linearly with the quantity of a substance; thus, integrated peak areas were calibrated to nmol using pre-determined HPLC-system-specific linear regression coefficients (Table S3). If no specific regression coefficients were available, we

applied the linear relationships of β-carotene for carotenoids and the one of chlorophyll *a* for green pigments. Most pigments discussed here were quantified by integrating Gaussian-fitted peaks on the chromatogram. Some exceptions are isorenieratene, car-52.58, lutein, zeaxanthin, α-carotene, β-carotene, and γ-carotene, which overlap significantly. The peaks of these pigments were integrated using a perpendicular drop method (Westerberg, 1969), which results in a higher error.

We also used a non-target approach and named unknown carotenoids according to their retention times (e.g. car-21.2 occurs

at 21.2 minutes after injection). We classified green pigments more polar than Chl *a* as pheophorbides-*x.x.x*; green pigments less polar than Chl *a* as pheophytins-*x*; and green pigments less polar than β-carotene as pyropheophytins-*x* (Lami et al., 2009). We are aware that this classification may be incorrect in some specific cases. For example, pheophorbides between 30-45 min could also be bacteriochlorophyll *c, d* homologues (Romero-Viana et al., 2010).

Pigments were interpreted using the interpretation keys of Bianchi and Canuel (2011), Schlüter et al. (2006), Lami et al. (2000)

and the carotenoid database Yabuzaki (2017) (Table S3). The chlorophyll preservation index (chl *a* / (chl *a* + pheophytin *a*)) (CPI) was used as an additional indicator of pigment preservation (Buchanan and Catalan, 2008). For some specific pigments, alternative interpretations were found in the literature; in these cases, references are provided.

### 3.5 Statistical analyses

Data exploration was performed using principal component analysis (PCA). To differentiate between statistically different

lithotypes, Ward's hierarchical clustering was applied to the high-resolution HSI and XRF data (cutoff at the Euclidean distance of 17), the lower resolution P, Mn and Fe, and the HPLC-pigment data sets. As XRF data was centred and log transformed and HSI data was z-scored, the data were normally distributed and could thus be compared in one plot. We verified PCA results and tested for non-linear dependencies by performing non-metric multidimensional scaling (NMDS) to get rid of the "horse-shoe effect" in PCA (Podani and Miklós, 2002).

Carotenoids and green pigments were pre-selected using a threshold approach. Pigments not continuously present throughout the core (>9 zero values, n = 59) were excluded from further statistical analysis. Throughout the core, 44 carotenoids passed this threshold criterion. Carotenoids were grouped using Ward's hierarchical clustering on the correlation matrix of carotenoids (Kramer and Siegel, 2019). Subdivisions were created when subordinate patterns were observed in multiple pigment time series or when certain subgroups included diagnostic pigments with a specific ecological interpretation. CONISS clustering of

the carotenoid dataset follows Grimm (1987); the number of significant clusters was determined using the broken stick model.



We performed redundancy analysis (RDA; Legendre, 2012) and Rate of Change RoC analysis (Mottl et al., 2021) on the pigment dataset to determine which independent variables could explain the pigment variance and when the community significantly changed. To test whether anoxic phases had an irreversible effect on the algal community, we analysed the trajectories of the pigment community data in the eigenvalue space (second and third principal component scores). The Matlab

and R codes are available on GitHub (https://github.com/SJSchouten).

## 4 Results and interpretation

### 4.1 Composite core and chronology

The composite core AMS22-COMP1 is 8.74 meters long (Fig. S1). Basal gravel to sandy clastic sediment (8.74–7.65 m) is overlain by 2.65 meters of laminated inorganic silty clay (7.65–5.56 m), followed by 5.56 meters of gyttja (5.56–0.0 m). Here,

we focus on the lower core sections 5.7–4.8 m, which span the Late-Glacial and the transition to the Holocene.

The chronology (Fig. 2) is constrained by calibrated $^{14}$C AMS ages of nine taxonomically identified terrestrial plant macrofossils (Table S1), three pollen-based biostratigraphic markers (Fig. S4), and the Laacher See Tephra (13,006±9 yrs BP, Reinig et al., 2021), yielding consistent ages between 14.9 and 10.5 ka BP. The modelled age uncertainty ranges between 300 and 600 years. Despite continuous sediment sieving, no terrestrial macrofossils were found below 5.43 m sediment depth.

Thus, the sediment section >15 ka BP is constrained by the early regional *Betula* rise ~16±0.68 ka BP (Rey et al., 2020). Accordingly, we estimate the age of the transition between silty clay and fine detrital gyttja (5.55 m sediment depth) to ca. 16.2±1 ka BP. Ages for the initial lake formation in this region date to <19 ka BP (Rey et al., 2020), suggesting that 2.65 m of laminated silty clay at the bottom of the core was deposited in ca. 2-3 ka, yielding Sediment Accumulation Rate SAR of >1 mm yr$^{-1}$ in the lower part of the core. SAR between 16 and 10.5 ka BP were remarkably constant and an order of magnitude

lower (0.11 mm yr$^{-1}$).





**Fig. 2: True colour RGB picture and lithology (left) and age-depth relation of the composite core AMS22-COMP1 (right) of Late-Glacial Amsoldingersee sediments. Green violin plots indicate biostratigraphic tie points, whereas blue violins indicate the distribution of $^{14}C$ age probabilities. The LST is marked in red. Priors to the Bayesian run are plotted on the right side.**

## 4.2 Lithology and sediment composition

Diagnostic inorganic and organic proxies, PCA time series, and Ward's hierarchical and CONISS-constrained clustering of all selected XRF and HSI proxies are displayed in Fig. 3. The complete data set is shown in the biplot of Fig. 4 and Fig. S5. Constrained and unconstrained clustering yielded similar results (six distinct clusters) suggesting that changes in the sediment composition (lithotypes) are predominantly stratigraphically constrained.



The PC1 axis (65% of total variance; Fig. 4) separates organic sediment (aquatic primary production; positive) from litho-clastic sediments (erosion, runoff, negative). PC2 (21% of variance) reflects redox and lake mixing conditions, whereby positive PC2 scores represent high Bphe $a$ concentration (well-stratified lake with hypolimnetic anoxia), and negative PC2 scores relate to well-mixed oxygenated conditions with sequestration of redox-sensitive metals such as Mn and Cu. The PCA shows signs of non-linear dependency ('horseshoe effect'). Therefore, we performed standardized-euclidian-MDS (Fig. S6), already yielding a decent fit with just one component (at a stress of 0.14). On the two-component MDS (stress of 0.06), we performed Ward's clustering, which annotated clusters similar to those of the PCA; thus, we conclude that our PCA-derived clusters are robust.

*Lithotype 1* (LT1, 5.70 m–5.56 m, ~18/~16.2 ka BP; Heinrich Event 1 (H1)) is a light grey (7.5YR5/1 Munsell) inorganic (0.4% $C_{org}$ and 21% water content) homogenous detrital carbonaceous silty clay that contains no plant macrofossils or diatoms and is poorly sorted with most grains being sub-rounded to rounded. The lithotype is elevated in Ca, Sr, Fe, Si, Ti and TIC (4.5–6%). Pigment concentrations are very low because of low aquatic primary production (PP) and high accumulation rates of fine clastic sediments (matrix effect). We interpret this lithotype as eroded glacial till washed into a young oligotrophic perennial lake.

*Lithotype 2* (5.56–5.51 m, ~16.2–15.8 ka BP, H1) is a light brown (10YR4/1 Munsell) heterogenous organic silty clay (3.5% $C_{org}$ and 46% water content) and contains no plant macrofossils or diatoms, and angular to rounded grains. It is elevated in Fe, Ti, and Si. TIC (1%) and Ca, Sr decrease rapidly. The sediments are enriched in S, and C/N ratios <10 suggest a predominantly aquatic origin of the organic matter (Meyers et al., 1997). The increase in aquatic PP is marked by a moderate increase in green pigments (Tchl), an increase in phycocyanin (cyanobacteria) and high amounts of Bphe $a,$ suggesting a chemocline in the photic zone and hypolimnetic anoxia. This lithotype is interpreted as predominantly clastic oligotrophic lacustrine deposition in a more stable deglacial landscape with widespread lake stratification and anoxia, and lower SAR during H1 (Oldest Dryas).

*Lithotype 3* (5.51–5.38 m, 15.8–14.7 ka BP, H1) is a light brown (10YR3/2) laminated (1–10 mm) heterogenous detrital gyttja (3.5–8.5% $C_{org}$ and 48-60% water content, 0.1–1.5% TIC), with sparse plant macrofossils and diatoms, and subangular grains. The organic matter content fluctuates with the sediment brightness. Inorganic laminae contain more clastic sediments with sharp contacts and darkening upward (elevated Ca, Sr, Ti, Fe, and Si). Organic laminae contain higher concentrations of CNS, Tchl and phycocyanin; C/N values are <10. Bphe $a$ concentrations lie around the detection limit except for a distinct local peak around 5.44 m. Generally, LT3 is relatively enriched in Ti, Zr and Fe and depleted in Ca, Sr and TIC (top left quadrant in the PCA, Fig. 4) suggesting that there is a source of siliciclastic sediment (possibly airborne) that comes from outside the carbonate-dominated catchment. We interpret sediments of LT3 (the latest part of H1) as lacustrine deposits in a deglacial landscape, with reduced erosional influx (lower Ca and TIC), increasing aquatic PP and traces of airborne dust (high Ti and Zr).

*Lithotype 4* (LT4, 5.38–5.21 m, and 5.09–5.05 m, 14.7–12.9 ka BP and 11.9–11.7 ka BP, respectively; Bølling/Allerød (B/A) and last part of Younger Dryas (YD)) is a dark brown (10YR3/2 Munsell) homogeneous organic detrital gyttja (15–28% $C_{org}$ and 72-80% water content, 0.5% TIC) with *Betula* macrofossils, sparse diatoms and rounded grains. LT4 has low Ti, Fe, and





Si and high S and Mn values. C/N ratios are higher than before and range between 10 and 11. Tchl (aquatic PP), $Cr_{coh}Cr_{incoh}^{-1}$
and PC1 are moderately high. Bphe *a* (PSB, anoxia) concentrations are variable, with local peaks at around 5.31 and 5.24 m. Phycocyanin is constantly present, reaches a maximum at 5.23 m, and remains high afterwards. In the upper section of LT4 (5.09–5.05 m), all pigment concentrations decrease and Bphe *a* gets close to the limit of quantification. We interpret sediments of the lower part of LT4 (B/A) as deposits in a stable landscape with relatively lower clastic influx and relatively higher aquatic PP. The lake was well mixed with poor development of hypolimnetic anoxia except for two episodes at around 5.30 m and
5.22 m sediment depth.

*Lithotype 5* (LT5; 5.21–5.19 m and 5.18–5.09 m, 13.0–11.9 ka BP; Younger Dryas) is a brown (10YR2/2 Munsell) homogeneous detrital gyttja with slightly lower OM concentrations than in LT4 (15–25% $C_{org}$ and 71–78% water content), abundant diatoms and *Pinus* and *Betula* macrofossils. Grains are mostly sub-rounded and poorly sorted, and the lithotype contains low Ca, Sr and TIC (0.7%) but elevated Ti, Fe, Si, K, Zr and S values. This suggests that, very similar to LT2 and
LT3 (PC2 positive, Fig. 4), the clastic sediment fraction bears a significant proportion of allochthonous sources, potentially airborne mineral dust. In contrast, erosion of carbonates from the catchment remained limited. S concentrations remain high, which is different from the inorganic LT2. Phycocyanin is generally high and reaches two maxima (5.21–5.18 m and 5.14–5.10 m). Bphe *a* (PSB, anoxia) and Tchl (aquatic PP) values are highest throughout LT5. We interpret LT5 (from LST onwards, YD) as lacustrine sediments formed in a dry (low surface erosion in the catchment) and cold environment with high dust loads,
permanent hypolimnetic anoxia, and high aquatic PP.

*Lithotype 6* (LT6, 5.05–4.8 m, 11.7–11.3 ka BP; Early Holocene) is a dark brown (10YR2/1 Munsell), homogenous highly organic detrital gyttja (30–35% $C_{org}$ and 81-85% water content, 1% TIC) with reduced lithoclastics. This unit contains large terrestrial macrofossils, diatoms and sparse sub-rounded to rounded grains. C/N ratios are relatively high and range from 10 to 11, comparable to LT4; Tchl (aquatic PP) and phycocyanin (cyanobacteria) remain at levels similar to LT4 (B/A). Bphe *a*
values are below the limit of quantification, suggesting that the lake was well mixed. LT6 (early Holocene) was deposited in a stable environment with very low lithoclastic input and high aquatic PP.



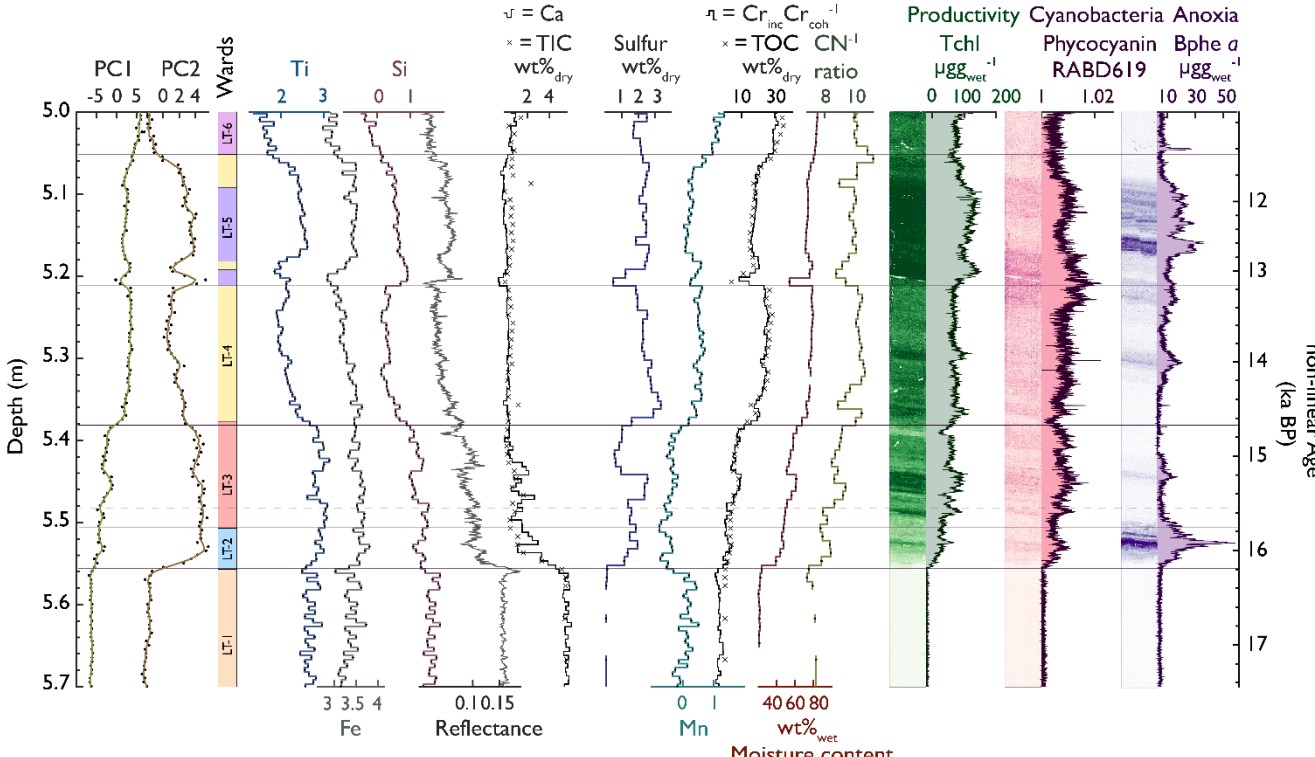

**Fig. 3: PC1 and PC2 values (data from Fig. 4), Ward's hierarchical unconstrained and CONISS constrained clustering (horizontal lines), and selected biogeochemical proxies. The hyperspectral data (right) are plotted as intensity maps and time series. The colour code of the LT is the same as in Fig. 4.**





**Fig. 4: Principal component plot of the XRF and HSI proxies. Ward's clusters are coloured along their confidence ellipses. The grey band indicates a smoothed depiction of the standard deviation of the PC1 and PC2 data, providing a band of 'common occurrence'. The environmental interpretation of the PCs is indicated in coloured boxes.**



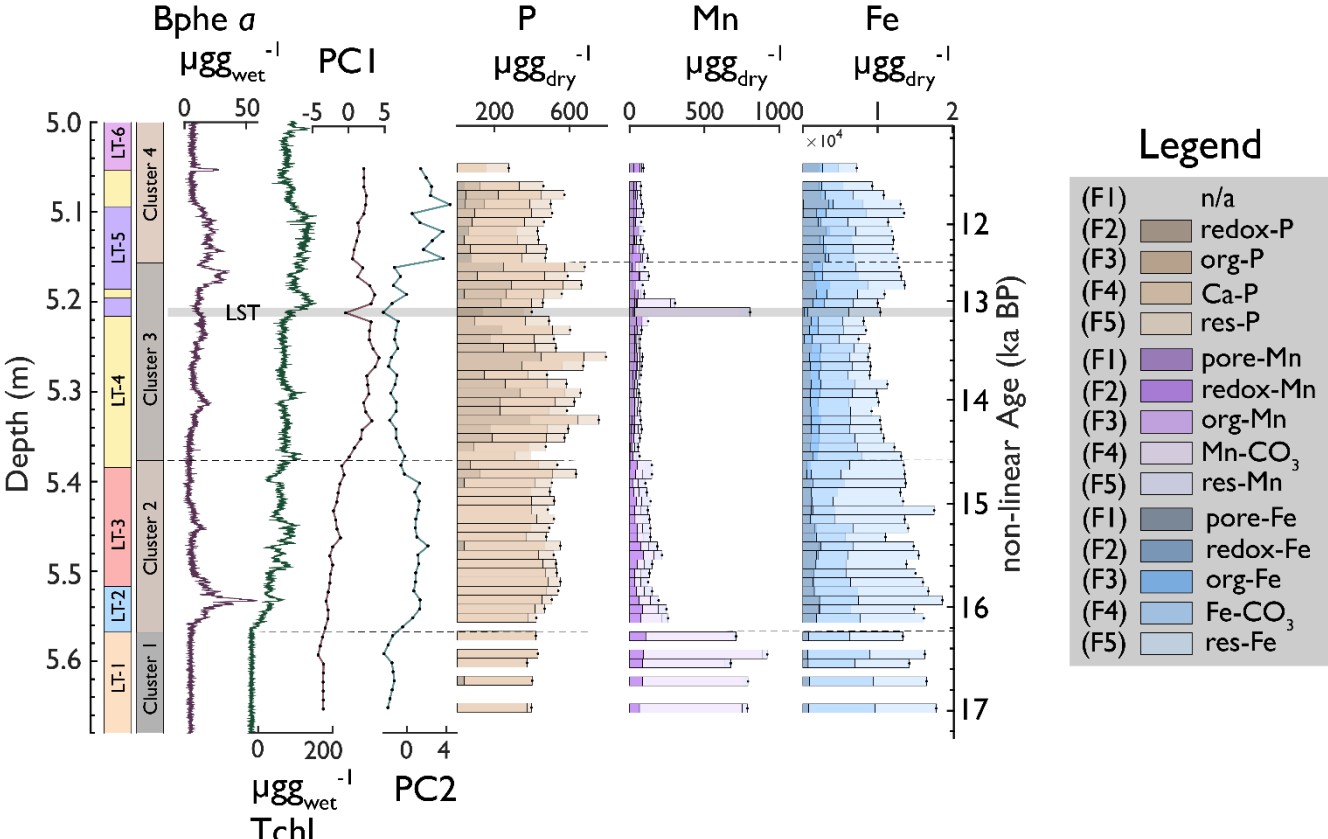

**Fig. 5: Down-core stacked bar plots of P, Mn, Fe fractions, PC1,2 and Clusters 1-4 of P, Mn, Fe fractions (data from Fig. S7). Hyperspectral proxies and lithotypes are shown for orientation on the left. Darker colours indicate the redox-sensitive and readily dissolvable fractions of P, Mn and Fe.**

### 4.3 Fractions of P, Mn, and Fe

CONISS and Ward's clustering of Fe, Mn and P fractions revealed four significant clusters (Fig. 5 & S7) with transitions that align with the lithotype changes. The four clusters are ordered along PC1 (48% of total variance), which goes from residual (lithogenic) fractions (PC1 negative) to organic fractions (PC1 positive), and PC2 (19% of total variance) goes along redox conditions (anoxia PC2 positive). The trajectories (Fig. S7, grey arrows) display a trend from catchment processes (16–15 cal. ka BP) with Ca-bound and residual P (F4 and F5) and Fe and Mn oxides, oxyhydroxides and carbonates (F4) and residual Fe and Mn (F5), towards organic-bound fractions (F3, 14–13 ka BP) and elevated redox-sensitive fractions (13-11 ka BP).

Total phosphorous is relatively stable throughout the record ($520 \pm 94$ µg g$^{-1}_{dry}$). Ca-bound phosphorous (P-F4) is the dominant fraction, ranging from 31–89% (mean = 63%) in the lower and upper parts of the core (below 5.37 m and above 5.18 m, corresponding to H1 and YD, respectively), whereas the Ca-P fraction is relatively small (28–55%) in the middle part of the core (B/A). Interestingly, Ca-P covaries with elements attributed to allochthonous siliciclastics from atmospheric dust (Fig.





3B, like Ti $r^2$ = 0.92, p < 0.001) and correlates less with elements interpreted as clastic sediments from local runoff like Ca ($r^2$ = 0.54, p < 0.001), suggesting that significant proportions of Ca-P, and thus P, are related to allochthonous inorganic dust deposition during cold periods (H1 and YD). The fraction of residual phosphorous (P-F5) is relatively small (mean = 17%) and stable throughout the record.

During the Bølling/Allerød (5.37–5.18 m), P-org (F3) covaries with sediment organic matter TOC ($r^2$ = 0.7, p < 0.001). Redox-330 sensitive P (P-F2) was undetectable in almost all samples throughout the record, suggesting that bioavailable P was minimal and, if present at all, constantly internally recycled, particularly during anoxic periods.

In contrast to P, total Mn varies between 100 and 900 µg g$^{-1}_{dry}$. Residual Mn is generally low (13–61 µg g$^{-1}_{dry}$) and covaries with Ti, Zr. Ca-bound Mn (Mn-F4) and redox-sensitive Mn (Mn-F2) have the largest variability. Carbonate-bound Mn occurred in the silty clay of LT-1 and the LST. TIC (carbonate), Ca, Sr, and Mn and Fe carbonates have a similar downcore 335 distribution and, likely, have a common origin. Porewater and organic Mn and Fe (Mn-F1, Fe-F1 and Mn-F3, Fe-F3) occurred in the most organic sections of the record (LT3 and LT4: 5.37–5.05 m) and covary with $Cr_{inc}Cr_{coh}^{-1}$ and TOC.

The redox-sensitive fraction of Mn was abundant during the H1 and YD phases (< 5.37 m and 518–5.05 m). Under conditions of hypolimnetic anoxia (high Bphe $a$), redox-sensitive Mn (Mn-F2) decreased, suggesting partly reductive dissolution and recycling. Residual Fe-F5 is more abundant than Mn-F5 and mostly elevated during the H1 and Younger Dryas phases (< 340 5.37cm and 5.18–5.05 m). F4 and total iron concentrations were more stable (800–1800 µgg$^{-1}_{dry}$).

## 4.4 Pigment stratigraphy

We identified four significant carotenoid groups (Fig. 6). Group 1: anoxygenic phototrophic bacteria (APB) peak during the early phases of H1, Older and Younger Dryas. Group 2: APBs, diatoms and other silicifiers peak during H1 and the YD. Group 345 3: purple non-sulphur bacteria, green algae, dinoflagellates and euglinids follow annual temperatures and peak in the Bølling. Group 4: mainly cryptophytes and cyanobacteria are constantly high throughout the Bølling, Allerød and the YD.

CONISS clustering revealed five pigment zones (PZ, Fig. 6 and 7) which correspond well to the Lithotypes (Fig. 3). These were also the phases of significant changes in algal composition as highlighted by RoC analysis: RoC maxima (Fig. 7) occurred after the LST at the transition to the YD suggesting that these two events had the largest impact on primary producer 350 communities. Further phases of change (insignificant) happened mainly during Heinrich-1 and during periods of rapid warming (onsets of Bølling and Holocene).





**Fig. 6: Down-core concentrations of pigments and HSI together with a heatmap of grouped carotenoids. Left, sums of Chl *a* and diagenetic products (stacked bar graph), HSI indices (for orientation) and CPI. Middle: Heatmap of the z-scores of carotenoids ordered to their statistical correlation into four carotenoid groups (bottom dendrogram) with subdivisions. Right: Pigment zones derived from CONISS clustering with subzones (arbitrary, horizontal dashed lines) and total carotenoid sums (shaded). Pigment concentrations are shown in Fig. S8.**

Pigment Zone PZ-1 (below 5.56 m; <16.1 ka BP, corresponding to LT1) contains low amounts of pigments, primarily representing diatoms (Group 2.2) and APB (Group 1), suggesting oligotrophic conditions in a young deglacial landscape. PZ-2 (5.56–5.38 m, 16.1–14.7 ka BP, corresponding to LT2 and 3) marks an increase in aquatic production (chlorophylls and carotenoids) and shows a pigment succession with 3 phases (PZ2a, b and c, Fig.6): initially, APBs (Group 1) dominate with admixtures of carotenoids related to diatoms and other silicifiers (Group 2) and traces of cyanobacteria (Group 4.3). This pigment assemblage suggests intense and prolonged lake stratification or even meromixis with a chemocline in the photic




zone. A minor anoxic phase is also observed later in PZ-2 at around 5.45 m (15.2 ka BP). Subsequently (5.49-5.44 m; PZ-2b), the APB community (Group 1) mostly disappears and is replaced by purple non-sulphur bacteria PNSB (Group 3.1), green algae (Group 3.2) and some cyanobacteria (Group 4.3). PZ-2 ends with a relative decrease in total carotenoids except for some specific nutrient-stress-related carotenoids of green algae (Group 4.1); this phase has a low chlorophyll preservation index
(CPI), suggesting pigment degradation related to high water clarity and/or shallower water.

PZ-3 (5.38–5.21 m; 14.7–13.1 ka BP, corresponding to LT4) shows a further increase in aquatic PP (total chlorophylls and diagenetic products, total carotenoids). Pigment assemblages are generally diverse, representing purple non-sulphur bacteria PNSB, green algae, euglenids, (stressed) green algae (Table S3), cryptophytes and cyanobacteria (Groups 3 and 4). Except for the short anoxia event at 5.32 m, APBs are hardly present (Groups 1 and 2.1). At the onset of PZ-3 (5.38 m; 14.7 ka BP; start
of Bølling warming), we observe a second succession (PZ-3a; Fig. 6) from astaxanthin (Group 4.1) to chloroxanthin (Group 3.1) to diatoxanthin (Group 2.2) to lutein (Group 3.2) and echinenone (Group 4.3). This suggests a successive shift from nutrient-stressed via nutrient fixing to more favourable growing conditions with abundant pigments and high pigment diversity (PZ-3b). This succession takes ca. 800 years after the onset of the warming. During and after this succession, pheophorbide *a* concentrations increase (absolute and relative to chlorophyll *a*), suggesting simultaneous zooplankton grazing (Table S3).
Towards the end of PZ-3 (PZ-3c), a relatively minor community shift occurs from green algae towards more cyanobacteria. PZ-3 ends with the LST.

PZ-4 (5.20–5.10 m; 13.07–11.90 ka BP, corresponding to LT4 and 5) spans the late Allerød and the early YD. Immediately after the LST, the community is dominated by diatoms, other silicifiers, and cryptophytes/dinoflagellates (Groups 2.2 and 4.2), suggesting silica fertilisation through tephra deposition. Tchl (HSI) and total carotenoids reach maximum levels. Our data
shows that the tephra fertilisation effect lasted for several decades. Diatoms and cryptophytes remain abundant in the second part of PZ-4 (corresponding to the YD) (Groups 2.2 and 4.2). Bphe *a* (APB) and carotenoids related to anoxygenic bacteria (Groups 1 and 2.1) return, suggesting strong lake stratification, prolonged hypolimnetic anoxia and a chemocline in the photic zone. PZ-4 also includes many pigments typically produced by (filamentous) cyanobacteria (Groups 4.2, 4.3, and 4.4).

PZ-5 (5.10–5.01 m, 11.90–11.32 ka BP) corresponds to the end of the YD (LT-4) and the Early Holocene (LT-6). During the
end of the YD (PZ-5a), all pigment concentrations are relatively low and primarily represent cyanobacteria (Group 4.3), non-sulphur bacteria and diatoms (Group 2.1). This drop corresponds to a phase with low CPI values and is possibly linked to enhanced uniform photodegradation. PZ-5b corresponds to the onset of the Holocene, with higher aquatic production dominated by green algae and subordinate cryptophytes, cyanobacteria, and purple non-sulphur bacteria. Group 3.2 gains immediate dominance following the abrupt Holocene warming. The lack of algal succession suggests that nutrient constraints
were less critical at the Holocene's onset than the Bølling's onset. After the onset of the Holocene, pigments are better preserved (increasing CPI).

To observe pigment composition in multivariate space, all extracted pigments (carotenoids and photopigments) are displayed as variables to their second and third principal components in Fig. 8a. PC2 (9% of the variance) and PC3 (7% of the variance) discriminate along taxonomic and ecological communities. In comparison, PC1 (63% of the variance) follows total



productivity (e.g. Tchl) and is, therefore, not shown. PC2 separates pigments produced during anoxic, cold and dusty phases (positive) from pigments produced during oxic, warm, nutrient-limited phases (negative). PC3 separates (pyro)-pheophytins, indicating photodegradation of chlorophylls from pheophorbides, indicating enzymatic breakdown of chlorophylls, including zooplankton grazing (Bianchi and Canuel, 2011; Lami et al., 2000). PC3 also provides a distinction between diatoms and PSB. Clustered pigments matching the carotenoid groups are likely produced by the same organisms or in similar environments.

Four non-bacterial photopigments (ppb-48.2, py-51.9, py-59.9 and ppb-49.6) occur in the quadrant with anoxic carotenoids. These substances could be homologues of bacteriochlorophyll *b,c,d,f* (Romero-Viana et al., 2010) produced by green pigment-producing APBs.

We analysed pigment-PCA scores (PC2 and PC3) in a trajectory plot (Fig. 8b) to test if Late-Glacial anoxia invoked irreversible changes in the algal composition. Anoxygenic phototrophs dominate the initial algal community. The community develops

towards green algae and cyanobacteria during the B/A interstadial, transitions to diatoms during the YD and, finally, ends close to the starting point, suggesting that the overall Late-Glacial trajectory (dashed arrow) left no irreversible compositional imprint. Also, the trajectories through the three main individual anoxic phases (Figs. 6 and 8b, purple, red and blue) returned to a comparable state concerning PC2 (taxonomy and ecology). In contrast, individual anoxia trajectories ended up with different PC3 values (degree of degradation), suggesting that, at the end of anoxia event 2, grazing pressure was higher than

at the beginning, whereas for anoxic phases 1 and 4, grazing pressure seems to be lower.





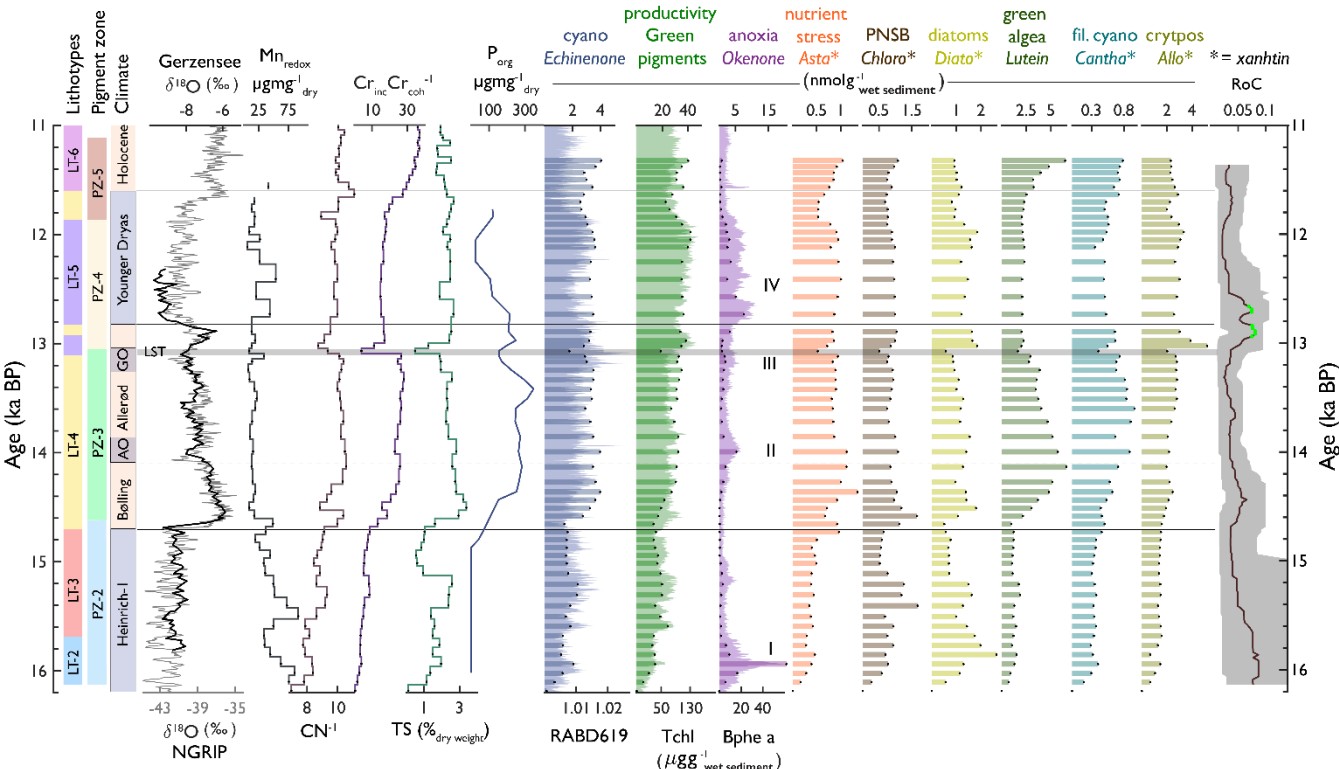

**Fig. 7: Compilation of Lithotypes, Pigment Zones, chronostratigraphic zones (with δ18O of Gerzensee and NGRIP to illustrate temperature, Ammann et al. 2013), selected geochemical proxies (Mn_redox, C/N, Cr_coh Cr_incoh^-1, TS and P), selected pigments and Rate of Change (of HPLC pigments).**

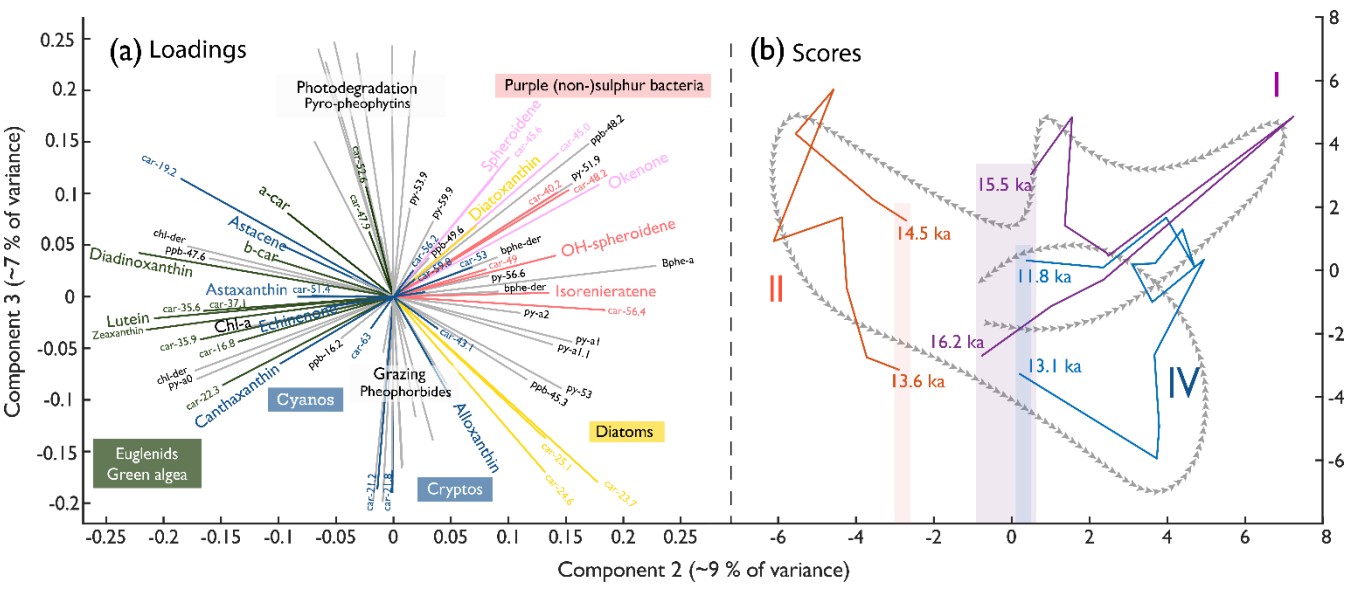



**Fig. 8: Principal component loadings (left) separating the algal communities and score trajectories (right) of all extractable pigments. (a) PC2 and PC3 of the pigment data. Carotenoids are coloured according to their statistical grouping (Fig. 6); Photopigments: ppb = pheophorbides; ppy = pyropheophytins; py = pheophytins. (b) Trajectories of the PC2 (9% of variance) and PC3 (7% of variance) scores throughout the three major anoxic phases (I, II, and IV; Fig. 7); shadings indicate starting and ending point range of the individual anoxic phases along PC2. Grey arrows indicate a schematic of the overall trajectory during the Late-Glacial, e.g. the same as for I, II and IV but then covering the whole record.**

## 5 Discussion

### 5.1 Late-Glacial lake evolution in the context of climatic and environmental conditions

In Amsoldingersee, sediment composition reflects aquatic organic matter vs. inorganic matter (PC1 axis, Fig. 4) and two different sources of clastic material, aeolian dust and runoff from the catchment (PC2, Fig. 4). Pigment assemblages are influenced by temperature, lake stratification and oxygen availability, nutrients and degradation processes. In our core (AMS22-COMP1), the lithotypes and pigment zones align stratigraphically (Fig. 7), suggesting that the processes controlling sediment composition and pigment assemblages have a common driver.

Earlier studies from nearby Gerzensee (Eicher, 1987) and other Swiss sites (Lotter et al. 1992) show that environmental changes on the Swiss Plateau responded in greatest detail to large-scale climatic shifts in the North Atlantic domain during the Late-Glacial (Ammann et al., 2013; van Raden et al., 2013). To place our findings in context with the large-scale Late-Glacial paleoclimate evolution, we discuss the development of Amsoldingersee along chronozones (Ammann et al., 2013).





Fig. 9: Comparison of proxy time series from Amsoldingersee (this study) with sites in the North-Atlantic/European domain between 16.5 and 11 ka BP. (a) Bphe *a*, anoxia (this study); (b) Tchl, primary production (this study); (c) Pollen records from Amsoldingersee (this study) and Moossee (Rey et al., 2020); (d) Meerfelder maar varve thickness (Brauer et al., 2008); (e) Kråkenes TiK representing runoff (Bakke et al., 2009); (f) Sieben Hängste speleothem δ$^{18}$O, with higher values representing more northerly flow (Luetscher et al., 2015); (g) log of NGRIP dust counts, and Ti abundance (this study); (h) Atmospheric CO$_2$ (Köhler et al., 2017); (i) Chironomid-inferred July temperature from the Alpine stack (blue, Heiri et al., 2015) and Burgäschisee (red, Bolland et al., 2020); (j) δ$^{18}$O of Gerzensee calcite (van Raden et al., 2013) and NGRIP (NGRIP members, 2004); (k) Insolation curves for July (red) and December (blue). Lines indicate synchronous time horizons across sites (shown at the bottom). Bottom: chronozones, AO = Aegelsee Oscillation (GI-1d) and GZ = Gerzensee Oscillation (GI-1b).





### 5.1.1 Heinrich Stadial 1 (H1; ~18 - 14.69 ka BP)

Amsoldingersee formed during H1, when retreating glaciers left a depression in the drumlin landscape (Fig. 1a). The sediment record starts with 2.65 m of laminated calcareous silty clay deposited into an early-deglacial oligotrophic perennial lake. Around 16.2 ka BP, the sediment composition shifts to an organic gyttja with a sharp decrease in sedimentation rate accompanied by an increase in organic matter deposition and a substantial reduction in detrital clastic input. Although no changes in local temperature and vegetation composition are observed, the shift suggests increased landscape stability and is concomitant with rising atmospheric $CO_2$ (Köhler et al., 2017; Fig. 9H).

After 16.2 ka BP, the composition of clastic sediments in Lake Amsoldingen changed (PC2, Fig. 4): Ca, Sr and TIC sharply dropped, whereas Ti, K, Si, and Zr remain at high levels, suggesting that there are two different sources of clastic sediments: one carbonate-dominated fraction (PC2 negative) related to erosion of calcareous glacial deposits in the lake catchment (low in siliciclastics) and another fraction of siliciclastic material (PC2 positive) related to allochthonous aeolian dust. The elevated and isolated topography of the Amsoldingersee catchment excludes fluvial transport as a potential source of siliciclastics (Fig. 1a).

The interpretation of the Ti, Si and Zr records in Amsoldingersee as aeolian dust is supported by the similarity of Ti to NGRIP dust (NGRIP members 2004, $R^2_{adj} = 0.8$, Fig. 9G) showing relatively high values during H1 and the Younger Dryas (YD) and relatively lower values during the Bølling/Allerød and the Early Holocene. Local dust maxima are observed in the NGRIP and Amsoldingen sediments also synchronously during the cold episodes of the Aegelsee Oscillation (AO; GI-1d, 14.1 ka BP) and the Gerzensee Oscillation (GZ; GI-1b, 13.2 ka BP). Whereas NGRIP dust is mainly sourced from Europe and Central Asia (Újvári et al., 2015), a large part of the dust deposited across Europe was sourced from sparsely vegetated steppes of the English Channel and the North Sea (Rousseau et al., 2014). The similarity of the Amsoldingersee and the NGRIP dust records supports the view that the same large-scale atmospheric circulation patterns affected the Swiss Plateau and Greenland (Eicher, 1987; Lotter et al., 1992, Ammann et al., 2013).

Across Europe, aeolian deposits are well documented for the H1, the AO (GI-1d), and Younger Dryas (Hoek et al., 2017). Estimated sedimentation rates for loess deposition range between 0.02 and 0.2 mm yr$^{-1}$ during Heinrich events across European sites (Rousseau et al., 2021). Given the lithological composition of the catchment with a predominance of calcareous glacial deposits over silicate-bearing lithologies and the low sedimentation rates in Amsoldingersee (SAR of 0.1 mm yr$^{-1}$; Fig. 2), dust fluxes on the above order of magnitude should be visible. This is quite particular for a lake sediment record. Within Switzerland, the "Etang de la Gruère" peat bog record is, to date, the only site that describes dust fluxes in the Late-Glacial. Similarly, Ti has been used as a dust proxy (Shotyk et al., 2001). Whereas the "Etang de Gruère" record covers the Holocene (0–14.6 ka BP), the Amsoldingersee record is, to our knowledge, the first continuous record of aeolian dust covering the Late-Glacial period (16–11 ka BP).

During H1, starting at 16.2 ka BP (PZ-1; Fig. 6), anoxygenic phototrophic bacteria (APB) and some cyanobacteria inhabited the lake, indicating hypolimnetic anoxia during the very early phase of the lake development. APB have an advantage over





other photosynthetic organisms in light-limited conditions (Karr et al., 2003); in Amsoldingersee, light could have been limited because of turbid water in deglacial windy environments and/or prolonged winter ice cover. Our results confirm earlier findings

of pre-Bølling anoxia in Lobsigersee (Züllig, 1986). Indeed, long ice cover favours lake stratification and anoxia (Klanten et al., 2023), even in oligotrophic conditions.

Around 15.6 ka BP and onwards (PZ-2a), aquatic PP increased. Summer season temperatures warmed by about 2°C as reported for the Greater Swiss Plateau (Rey et al., 2020; Bolland et al., 2020; Fig. 9I). This warming is not found in the $\delta^{18}O$ NGRIP record. Apparently, the summer warming around 15.8 ka BP initiated a simultaneous increase in terrestrial (*Betula,* Fig. 9C;

Rey et al., 2020; Fig) and aquatic production (mainly green algae and cyanobacteria; lutein and echinenone).

Whereas this regional summer warming persisted (Bolland et al., 2020), aquatic PP dropped in Amsoldingersee after 15.1 ka BP (PZ-2c Fig. 9A), and dust input increased (Ti, Fig. 9G). A trend towards less negative $\delta^{18}O$ values from the nearby Sieben Hängste speleothem (Fig. 8) has been interpreted as strengthened northerly flow during that time (Luetscher et al., 2015). The CPI is low, and pyropheophytins are relatively abundant, suggesting extensive photodegradation of pigments (Buchaca &

Catalan, 2008). Persistent drought and low lake levels could explain photodegradation during a cold phase within H1. Glacial readvances were documented in the Swiss Alps during that time (Daun, Clavadel/Senders, Ivy-Ochs et al., 2006).

### 5.1.2 Bølling/Allerød (14.69 - 12.90 ka BP)

In the North-Atlantic domain, the onset of the Bølling (start of DOE-1) shows a very sharp increase in $\delta^{18}O$ values in Greenland ice cores (Rasmussen et al. 2014). *Juniper* expanded rapidly on the Swiss Plateau, indicating an immediate response of

terrestrial plants to the rapid warming that occurred within decades (Lotter et al. 1992). Amsoldingersee became only slightly more productive. Total aquatic production follows chironomid-inferred July temperatures with positive trends rather than the saw-tooth structure of annual temperatures as shown by the NGRIP $\delta^{18}O$ or $\delta^{18}O_{CaCO3}$ records from Gerzensee (Lotter et al., 2012; Fig. 9J), suggesting that aquatic production mostly responded to warm-season temperatures. Whilst this is true for most algal groups, APB thrive during the short cold and dusty intervals of the Aegelsee and Gerzensee Oscillations (GI-1d and b)

when winter ice cover was prolonged, and hypolimnetic anoxia prevailed (Figs. 7-9). This supports the interpretation of both cold intervals being predominantly a phenomenon of the winter season cooling that had little impact on summer temperatures (Lotter et al., 2012; Ammann et al., 2013). Apart from these cold intervals, we do not find any indication of lake stratification and hypolimnetic anoxia during the Bølling/Allerød warm phases. This finding is surprising considering warm summer temperatures and closed forest cover (Arboreal Pollen >80%, Lotter et al. 1992), which typically leads to lake stratification

and anoxia (Zander et al., 2021). Seemingly, summer thermal stratification and aquatic PP were not strong enough to spark summer anoxia.

Significant blooms of silicifying algal groups are noted after the LST deposition (Figs. 6 and 9), suggesting a Si fertilisation effect of the tephra in the oligotrophic and possibly Si-limited lake. This effect lasted <100 years and was also observed in oligotrophic Moossee and Lobsigensee on the Swiss Plateau (Makri et al., 2020; Züllig, 1986).





### 5.1.3 Younger Dryas (12.90 - 11.7 ka BP)

The Younger Dryas (YD) was a cold phase in the North-Atlantic region (Brauer et al., 2008) that was accompanied by reduced Atlantic Meridional Overturning Circulation AMOC (Böhm et al., 2015) and enhanced insolation-driven seasonality in this region (Fig. 9K). In Lobsigensee (Fig. 1b), Züllig (1986) observed extensive winter anoxia, which he related to extended ice cover during very cold winters. Although summer temperatures decreased markedly during the YD (Heiri et al., 2015), total pigment concentrations remained high in Amsoldingersee, even after the fertilising effect of the LST disappeared, suggesting that summer temperature was no longer a controlling factor for algal growth.

The strong response of silicifiers to the LST suggests that the system was Si-limited at the end of the Allerød (before the LST). At the beginning of the YD, green algae (lutein, Fig. 7) decreased, whereas silicifiers (alloxanthin, diatoxanthin) remained high, suggesting that Si fertilisation related to persistently high dust fluxes (Ti, Fig. 9G) played a role. It is unlikely that enhanced P recycling under anoxic conditions stimulated aquatic PP during that time because redox-sensitive P, Fe and Mn were sequestered rather than released (Fig. 5).

In Amsoldingersee, during the YD, we find multiple changes in productivity and anoxia, suggesting that the YD was not a homogenous cold phase but had an internal structure (Pigati and Springer, 2022; Weber et al., 2020). The YD started with ~185 years of low Tchl and high Bphe *a* (Fig. 9A, B), interpreted as a period with reduced epilimnetic production, short summers and long-lasting winter stratification, similar to the findings of Brauer et al. (2008) in Meerfelder Maar. During the second part of the YD, productivity and anoxia covaried, and sequestration of redox-sensitive Fe, Mn, and P increased. The end of the YD was associated with a drought-induced lake-level lowering (Lotter & Boucherle, 1984). Dry conditions at the end of the YD were also found in other European sites (Weber et al., 2020; Brauer et al., 2008), consistent with the idea of gradually northwards shifting storm tracks throughout the Younger Dryas (Bakke et al., 2009, Fig. 9E).

### 5.2 What drove the algal communities?

Overall, algal community changes are synchronous with lithotypes and phases with high RoC (Fig. 7), which match the chrono- and biostratigraphic zones reflecting large-scale climatic variations in the North Atlantic domain (Ammann et al., 2013). Aquatic primary production in Amsoldingersee (e.g., Tchl, total carotenoids) covaries with TOC and follows predominantly, but not exclusively, temperature and arboreal pollen (Fig. S9). Our data reveals that nutrients and lake mixing also played an essential role in shaping primary producer communities. Several lines of evidence suggest that P, N, and Si in the photic zone were limiting aquatic PP, although they have changed relative importance over time.

Firstly, there is very little phosphorous sequestered throughout the entire record. In contrast to Soppensee (Tu et al., 2021), bioavailable P is barely present in Amsoldingersee (Fig. 5) despite well-mixed and oxic conditions during the more productive Bølling/Allerød, suggesting that P might have been particularly limited. Successions between algal groups provide a second line of evidence for the limitation of nutrients (P, possibly also N). Whereas the climate warming at the beginning of the Bølling was very rapid and instantaneous (Bolland et al., 2020; Heiri et al., 2015; van Raden et al., 2013; Lotter et al., 2012),





the response of the pigments (aquatic communities) was gradual, following a succession across ca. 500 years from astaxanthin (end of PZ-2c, Fig. 7) to chloroxanthin (PNSB), diatoxanthin (diatoms), echinenone (cyanobacteria) and to lutein (green algae). This suggests that, at the onset of the Bølling, primary producer communities possibly did not predominantly respond to temperature but rather to nutrient availability. The timing of the algal succession is matched by successive increases in redox-Mn, C/N, S and eventually $P_{org}$ (Fig. 7).

Secondly, nitrogen was likely limited during H1 and the onset of the Bølling. At the onset of H1, carotenoids associated with filamentous cyanobacteria (N-fixing, Thiel et al., 2004) contribute considerably to the total carotenoids. Moreover, phycocyanin and Tchl covary suggesting that a large share of chlorophyll *a* (Tchl) was produced by cyanobacteria. As mentioned above, the algal response to the Bølling warming was relatively gradual, starting with astaxanthin and chloroxanthin (PNSB). Astaxanthin is a UV-protective carotenoid that *Haematococcus pluvialis* can use to store nutrients, which has been interpreted as N-limitation and light stress (Orosa et al., 2001; Boussiba and Vonshak, 1991). Also, chloroxanthin is a pigment which is produced by purple non-sulphur bacteria of the *Rhodospirillum* or *Rhodobacter* genera, which are known nitrogen-fixers (Albrecht et al., 1997; Keskin et al., 2011). Filamentous N-fixing cyanobacteria (canthaxanthin and echinenone) remained abundant during the Allerød and YD, suggesting that cyanobacteria still had a competitive advantage in N-limited conditions.

Lastly, at the beginning of the Bølling, a shift from silicifiers (diatoxanthin) to green algae (lutein) suggests Si limitation. This coincides with a decrease in dust (Ti, Si) deposition (Fig. 9G). Also, the strong response of silicifyers to the LST (13.05-12.90 ka BP; Fig. 7) suggests a substantial Si-fertilization, whereas green algae (lutein and Chl *a* - related pigments) or cyanobacteria remained invariant. Blooms of diatoxanthin (diatoms) and alloxanthin (cryptophytes) persisted during the dusty YD. The synchronous decrease in aquatic PP and dust deposition at the end of the YD suggests that Si fertilisation persisted throughout the YD. It is unusual to find signs of silica fertilisation related to the LST (Züllig, 1986) or dust in lake sediment records because these typically contain substantial siliciclastic fractions (Koffman et al., 2021).

## 5.3 Ice-induced Anoxia in Late-Glacial Amsoldingersee

Amsoldingersee documents four phases with hypolimnetic anoxia; all of them are related to cold periods during H1, the Aegelsee Oscillation (GI-1d), the Gerzensee Oscillation (GI-1b) and the Younger Dryas. This is consistent with findings from Züllig (1986) in Lobsigensee (Fig. 1b), who documented anoxia during cold periods of the 'Oldest Dryas' (H1) and the Younger Dryas and related these to prolonged winter ice cover. Although still little is known about winter limnology (Jansen et al., 2021; Hampton et al., 2017), there is increasing evidence that extensive winter ice cover leads to anoxia in shallow lakes at high latitudes (Klanten et al., 2023). These conditions likely occurred in Switzerland during the cold phases of the Late-Glacial. Many studies from the Arctic and Antarctica (Schütte et al., 2016; Reuss et al., 2013; Comeau et al., 2012; Rogozin et al., 2009; Karr et al., 2003) show that APBs and PNSB are well adapted to light limitation under ice cover as they use bacteriochlorophyll *a* which absorbs light at 550–600 nm where transmittance trough ice is maximal (Bolsenga et al., 1991).





Pigments used by other organisms (e.g., Bchl *c, e, d* and Chl *a, b*) absorb light only at the edges of the light spectra transmitted through snow and ice (<480nm and >630nm; Zander et al., 2023; Stomp et al., 2007).

Our results and those of Züllig (1986) are in contrast with previous findings in Längsee (Schmidt et al., 2002) and Soppensee (Tu et al., 2021). Both lakes developed anoxia during the warm phases of the Late-Glacial and not during the cold phases. The authors proposed a chemical feedback model in which summer warming leads to high aquatic PP, lake stratification and anoxia, which prompts reductive dissolution of redox-sensitive metal oxides and recycling of nutrients, thus sustaining high aquatic

PP and hypolimnetic anoxia (Gächter et al., 1988; Nürnberg, 1998). The different behaviours of these four lakes are surprising because they are very similar in terms of climatic and environmental background conditions and show similar morphometric properties (size and depth). Our current understanding is too limited to provide a sound explanation for this discrepancy.

Amsoldingersee and Lobsigensee also developed anoxia at times of tundra vegetation (during H1) and reduced forest cover during the YD (Rey et al. 2020), whereas the lake remained well mixed during the Allerød with closed pine forest. This stands

in contrast to many small lakes across Europe that show anoxia during Holocene warm conditions. In such lakes, reduced wind mixing through closed forest covers (Arboreal Pollen >80%) prompted summer stratification with anoxia (Zander et al., 2021). On the contrary, in Amsoldingersee, the winter ice cover was more important than closed forest, prompting anoxia during Late-Glacial conditions. This phenomenon is, however, not sufficiently well understood.

## 6 Conclusions

This study provides a contextual view of the complex interactions between climate variability (warming and cooling), aquatic primary production and primary producer communities, lake mixing regimes, anoxia and nutrient cycling in a small glacial lake (Amsoldingersee) on the Swiss Plateau during Late-Glacial times. This period is characterised by rapid and high-amplitude climatic changes 18-11 ka BP, including Heinrich Stadial-1, the Bølling/Allerød, the Younger Dryas, and the onset of the Holocene. We hypothesise that the responses of lakes in early deglacial Switzerland can be used as analogies for modern

lakes in the Arctic under rapid warming.

Our record reveals four anoxic phases during Late-Glacial times; all of them occurred under cold periods with prolonged ice cover during H1 (16 ka BP), during the Aegelsee and Gerzensee Oscillations (GI-1d and b), and the YD. During the warm Bølling/Allerød, the lake remained well mixed. These results contrast with findings from other lakes (Switzerland, Austria), where anoxia occurred during warm periods of the Late-Glacial. This discrepancy is surprising since these lakes are in similar

climatic conditions and have similar morphometry, suggesting that very subtle differences may lead to diverse responses of similar lakes to the same climatic forcing. The reasons for this discrepancy are currently poorly understood, but they possibly reflect the very complex responses of small lakes to rapid warming as presently observed in the Arctic.

Changes in aquatic primary production and algal (pigment) communities are synchronous with changes in lithotypes, biostratigraphy in the catchment and large-scale climatic changes in the North Atlantic domain. Aquatic PP and algal

communities were driven mainly by summer temperature but strongly modulated by nutrient limitations (P, N, Si) throughout

Late-Glacial times. Anoxic phases changed the algal communities, but these shifts were fully reversible once anoxia vanished. Noteworthy are differences in the pigment preservation and grazing pressure before and after the anoxia. The first significant increase in aquatic PP is observed as early as in H1, around 16.1 ky BP, i.e. much earlier than the Bølling warming at 14.6 ky BP.

The sediments of Amsoldingersee provide a unique continuous record of dust deposition (Ti, Zr and Si) covering the complete Late-Glacial period (18-11 ka BP). This record very closely matches the NGRIP dust record, confirming earlier findings that both regions (Swiss Plateau and Greenland) are linked through large-scale climate phenomena. Similar to the strong Si-fertilizing effect of the Laacher See Tephra in Amsoldingersee, atmospheric dust also seems to have modulated the primary production of silicifying algae during the Bølling/Allerød and the YD.


*Data and code availability:* Pangea (PDI-39645, PDI-39644, PDI-39643; DOI coming soon…); code is available at GitHub (https://github.com/SJSchouten)

*Author contributions:* SS – conceptualisation, formal analysis and investigation, visualisation and code, writing of original draft.
PZ - conceptualisation, XRF data curation, formal analysis and investigation, co-supervision. NS – conceptualisation, formal analysis and investigation. AL – HPLC, pigment interpretation. PBK – macrofossil identification. JL – pollen analysis. HV – co-supervision. MG - conceptualised, supervised, interpreted, and contributed to writing the original draft. SS, PZ, MG, NS, and HV commented, SS, PZ, HV, and MG edited, and all co-authors approved the manuscript.

*Competing interests: None of the authors have competing interests.*

*Financial support:* Swiss National Science Foundation 200020_204220

*Acknowledgements:* We thank Willy Tanner for leading the coring campaign and the staff at the Institutes of Geography and
Geology for support in the lab. We thank Mr. Wolfgang Hegner for granting us access to the lake and Mr. Ernst for supporting us on the site. LANAT provided the research permission. We further thank Rik Tjallingii for his help processing XRF data and valuable comments on the manuscript.

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
