# Peer review of "Lake anoxia, primary production and algal community shifts in response to rapid climate changes during the Late-Glacial"

_EGUsphere, 2025_

## Referee Comment (RC1)

Review of Schouten et al Biogeosci, 2025

**Comment to author:**

Dear Stan and co-authors,

I very much enjoyed reading your manuscript entitled "Lake anoxia, primary production and algal community shifts in response to rapid climate changes during the Late-Glacial". I think the paper studies a topic that is of interest to palaeoecologists and palaeoclimatologists, and that is, as you hypothesise in your conclusions, relevant with respect to future climate warming particular in Arctic regions.

I think your manuscript is really well-written: it is logically organised and clearly explains your research questions, approach and results. The high-resolution results are presented clearly and the interpretations of the drivers of changes in the algal community, as well as the occurrence and impact of (ice-driven) anoxia, were very interesting to read.

I only have minor suggestions for improvements that I hope will help you when revising your manuscript – most of which are of a typographical nature. I'd like to compliment you on a really well-executed study that I think will be of interest to many in the field

Comments
L15: "a critical gap" – I would argue that lots of studies on lake ecosystem response to (LGIT) climate change are already available; of course we can always learn more (and we do in the remainder of your paper!), but I wouldn't call this a critical gap myself
L29: "a... sources" – singular vs plural
L53: could you explain how climate warming is reinforced by watershed dynamics?
L62: why is only one of the research methods highlighted here? Given that more information is provided on L75 onward I think this can be left out. Alternatively, perhaps a wider range of methods needs to be discussed on L62 as well
L75: you start with methods, then discuss the site, and then at L81 discuss methods again. I suggest to combine the methods either before or after the site introduction
L120: "continuously sieved" – suggest to change to "Consecutive (or contiguous) 1-cm-thick sediment samples were sieved". As it stands you are implying that you never stopped sieving
L126: Perhaps not for this paper, but why was no pollen counted for materials younger than the Allerød?
L212: Here it states RDA is carried out but it is unclear where this comes back later in the manuscript. The RoC results are reported clearly, but I didn't see the RDA explicitly reported on
L223: what does "consistent" mean in this context?
L224: add if this is 1 or 2 sigma
L271: Results 4.2: I find the description a bit confusing. I understand you go by LT rather than chronologically, but: LT4 is split up into two separate parts, with a larger unit during the BA and then a short unit at the end of the YD. LT4 (B/A) is given an interpretation, but the upper LT4 unit is not interpreted. LT5 is then split into two subunits in the "title" (L281) of 5.21-5.19 and 5.18-5.09m, but these subunits are sequential and the difference between these two parts of LT5 is never explained. I understand that in the Results section you are taking a LT by LT rather than a chronological approach,

but the reader would benefit from a clearer explanation of where the LT sequence differs from the chronological sequence

L316: "that align with" – this seems incorrect to me. Only the Cluster 1/2 transition aligns with a lithostratigraphic transition. Cl2/3 just post-dates the lithostratigraphic transition, and Cl3/4 is smack in the middle of LT5. It's actually very interesting to see that these indicators change at different points, so I would suggest to emphasise that

L350: I suggest to add "statistically insignificant" rather than just "insignificant" as you do report on and interpret these, so obviously there's some interest in this

L399: You do clearly show the importance of PC1, explaining a major 67% of the variability in your dataset. PC2 and PC3 are then only explaining very minor proportions at 9% and 7%, respectively. I would personally suggest rather than showing a bi-plot of PC2 vs PC3, to show one bi-plot of PC1 vs PC2, and another of PC1 vs PC3. There might be good reasons not to do this, but I think this would also visually represent the overriding importance of PC1, which is now somewhat lost

L461: could you explain how increasing $CO_2$ could lead to impact on the landscape or the lake in absence of changes in temperature and vegetation composition? Similarly, are there any (e.g. geomorphological) reasons for increased landscape stability in the absence of changes to the terrestrial vegetation?

L516: out of interest, is there a role for wind (strength) here? Is the lake currently experiencing strong winds due to e.g. tunnelling along the valley?

L532-539: Personally I find it very interesting to see the 3-stage development of the YD. As you well explain, for a long time the YD was seen as a stable event. Later, the 2-stage event interpretation got more attention following the papers by Bakke et al. (200g Nat Geosci) and Lane et al. (2013 Geology). More recently, there seem to be more studies appearing that show evidence of an even more dynamic event, including a 3-stage YD in the SE of the UK as shown by Francis et al. (in press, JQS) as well as further afield (Fastovich et al., 2022; QSR). The Weber et al. (2020) paper that you cite also shows additional complexity during the YD. Sorry this is more a note than a comment, but I do find this result really interesting!

L540: why is there no interpretation of the early Holocene lake environment? Surely it's interesting to the reader to know how the transition into the Preboreal happened, and if circumstances (e.g. Si limited) in the Preboreal were similar to those of the (late) A/B?

L540: "what drove communities" is a bit vague – do you mean community composition, overall productivity, or both?

Conclusions: The order of PP and anoxia is reversed in the Conclusions compared to the Discussions (where PP (section 5.2) precedes anoxia (Section 5.3); I suggest treating them in the same sequence in both the Discussion and in the Conclusions

---

## Referee Comment (RC2)

Comments on the paper by Schouten and co-authors entitled. *Lake anoxia, primary production and algal community shifts in response to rapid climate changes during the Late-Glacial.*

This paper investigates the sediments of a small lake on the Swiss plateau to reconstruct anoxia and algal communities from the Late-Glacial until the Early Holocene. The idea is to understand the response of a small lake to warming and its potential recovery afterwards. This kind of information is useful to better understand if the current warming has irreversible effects. The authors use numerous, complementary and cutting-edge methods to carry out a very high-resolution analysis of this 90 cm long sediment section. The methodology is well described, allowing to reproduce what has been done. The work is meticulous, and the analyses were carried out with rigour. A special congratulation for the use of the sequential extraction of P, Mn and Fe, something that should be done more often when it comes to interpreting these elemental profiles. Although the sedimentary section is very short, the amount of data produced is impressive. The presentation of this mass of data is sometimes difficult to follow, although the manuscript is generally pleasant to read. The right amount of information has been moved to supplements. Please note that I am not competent to assess the interpretation of pigment analyses by HPLC.

**General comments**

**Abstract**: the introduction to the topic is quite long and could shorten.

**Figures and call to figures.** While the figures are well drafted, it is often quite difficult to find the information described in the text.
For instance, the text of sections 4.2, 4.3 and 4.4 refers to the chronozones H1, Bølling, Allerød, YD and Holocene, but none of the figures called for (Figs. 3-6) contains that information. I understand that you have established three different stratigraphies, (litho-, cluster-, pigment-) and that you also use the chronozones, but the text is not easy to follow for the reader that is not familiar with the study.
I know that these figures are already very busy, but would it be possible to add these chronozones to Figs 3-6? It would make the paper easier to follow.
Along these lines, section 4.4 is hard to follow, because it is not clear what figure the text is referring to: is it 6 or is it 7?
To make things even more complex, you also use the NGRIP stratigraphy (e.g., (GI-1d)). I understand why you do that, but is it really necessary, since you are not really discussing the potential leads and lags of your lake with NGRIP?
Another example is at lines 492-493: one needs to switch from one figure to the other to follow your discussion. Here, you refer in the same sentence to PZ-2a, described in fig. 7, and to features outlined in fig. 9i.

In brief, please make sure that what is described in the text is easy to find in the figures.

**Calibrated dates**. All the dates have been calibrated using Intcal20, but none of the dates mentioned in the text and the figures indicate "cal". Please add "cal" before all occurrences of BP.

**Chronozones**. It is not clear how the chronozones have been defined. You write on lines 126-127: *We combined $^{14}C$ dates with well-dated regional biostratigraphic markers (Ammann et al., 2013; Rey et al., 2017, 2020)*), but I'm not sure to understand what you have done practically. Are you just relying on your age model, and use the dates of the chronozones defined by Ammann (or the NGRIP chronology) to set the boundaries used here? Or do you use the changes in one of your indicators to set the boundaries of the chronozones?
Along those lines, is your age model accurate enough to briefly discuss the timing of these chronozones? (This is probably another paper, but maybe somebody will want to do that.)

**Section 4.2.** At the end of the description of each lithofacies, there is an interpretation. I'm not against including interpretations of the lithofacies here, but one needs to make sure there is no repetition in the discussion section.

**Discussion**. Why the onset of the Holocene is not discussed? It was announced somehow in the introduction by saying that your study would help to understand the response of lacustrine systems in a quickly warming climate. Yet, the early Holocene is usually considered to have a very fast warming. If it is the recovery to warning that is of interest, please write it down in the text of the discussion.

**Section 5.5.1.** You are discussing AO, GZ, YD in this section dedicated to the description of H1. This is not logical. Maybe should you add a "dust record section" that covers multiple time slices.

**Other minor comments**
L47: one understands you are talking about species only later. I suggest writing : "(...) promote generalist species competitiveness (...)".
L105: I think you should specify that the DOE-1 corresponds to the Bølling because I'm not sure that the readership of Biogeosciences is familiar of the DOE stratigraphy.
L132-135: I'm not sure to understand what you have done here? You XRF scanned the core three times and measured the standard deviation of your measurements? Was this performed on clr-transformed data or on raw data?
L215: I have not seen the R codes on Github.
L228: here you should use kyr instead of ka, because it refers to some duration.
L237: only XRF data are shown in S5.
L249: how was water content measured? This is not in the method section.
L251: "*of low aquatic primary production (PP) and high accumulation rates of fine clastic sediments (matrix effect)."* belongs to a discussion section not a result section (see comment about section 4.2 in the general comments).
L301: please spell out what LT means in the caption.
L315: not all transitions align with lithotype changes.
L369-370: could the degradation of chlorophyll also be due to anoxia?

L413-415: what I understand from these lines is that the degree of degradation (PC3) is due to high grazing pressure during anoxic events. I don't understand this, as anoxia should prevent the presence of grazing organisms. Is there something I'm missing here?

L439: I suggest the following: "(…) *show that environmental changes on the Swiss Plateau responded **very closely** to large-scale climatic shifts (…)*".

L441: try to be more specific about how you have done that.

L458: I don't see anywhere a curve with sedimentation rates. This "sharp" decrease is not visible in the age model presented in Fig. 2. Actually, you have no date before 16.2 ka BP, and there is just the assumption that the lake formed around 19 ka. The sedimentation rate you infer is quite uncertain, even if it makes sense. I would be more cautious here.

L462-466: your interpretation makes a lot of sense, but you don't have the smoking gun. A stronger argument would be if you could show that the rounded grains in your sediment are Ti, K, Si and Zr-rich and the angular ones are from carbonate rocks.

L469: how this correlation has been calculated to cope with their different resolutions. Also, it is not easy to tell if the Ti curve in Fig 9g is the low resolution one or the opposite. More contrasted colours would help.

L481: I'm not sure this statement is true and useful.

L489: in my view, turbid waters are not compatible with low clastic sedimentation rates, or maybe there is something I don't understand here.

L497: this fluctuation is barely visible. Is it a question of scale?

L514: I do not see the 80% in this Bølling/Allerød interval. It starts near 0% and ends around 60%.

L537: "drought" is quite a strong word. In this small closed lake, a drought would have a more spectacular impact. Maybe should you simply say that the climate was drier, and that it produced a lowering of the lake level.

L541: one should expect high rates of changes between phases, but not during phases, right?

L568: Fig. 9g does not show Si.

L569: this idea of Si-fertilization has already been presented earlier. Try to avoid repetition

L571: what do you mean here? How the LST could Si-fertilize the lake before the eruption even occurred?

L618: replace the two occurrences of "ky" by "ka", as they are dates and not durations (and add "cal" of course).

I enjoyed reading this very interesting paper, and I hope these comments and questions will help to improve the manuscript.

Pierre Francus
INRS, Québec
18 March 2025

---

## Author Comment (AC1)

**Reply to referee 1**

We would like to thank reviewers for detailed and constructive reviews and comments. We appreciate the reviewers' thoughtful feedback and insights, which helped us improve the clarity and comprehensiveness of our work. We have revised all the points the reviewers suggested accordingly.

Our point-by-point responses can be found below:

*L15: "a critical gap" – I would argue that lots of studies on lake ecosystem response to (LGIT) climate change are already available; of course we can always learn more (and we do in the remainder of your paper!), but I wouldn't call this a critical gap myself.*

➔ **Replaced with** "While modern impacts are well-studied, knowledge on the responses of lake ecosystems to climate change in the pre-anthropogenic times is still sparse.".

*L29: "a… sources" – singular vs plural*

➔ **fixed**

*L53: could you explain how climate warming is reinforced by watershed dynamics?*

➔ **Sentence was edited to: "**It is increasingly recognised that rapid climate warming can affect nutrient loading to lakes through changes in the watershed dynamics, e.g., by affecting hydrological patterns, weathering rates and mass fluxes, and thus, is an essential driver of freshwater and ecosystem deterioration (Meerhoff et al., 2022; Jane et al., 2021; Jeppesen et al., 2010)."

*L62: why is only one of the research methods highlighted here? Given that more information is provided on L75 onward I think this can be left out. Alternatively, perhaps a wider range of methods needs to be discussed on L62 as well.*

➔ We removed the text highlighting the hyperspectral imaging and kept it only in the paragraphs below.

*L75: you start with methods, then discuss the site, and then at L81 discuss methods again. I suggest to combine the methods either before or after the site introduction.*

➔ Significantly adjusted the structure to implement these two comments: methods and site descriptions are now clearly separated.

**The text reads now as follows:** "To answer those questions, we investigate a sediment core from a small kettle hole lake on the Swiss Plateau, Amsoldingersee. This lake is located adjacent to Gerzensee in an area that responded in extraordinary detail to Late-Glacial climatic changes documented in the North Atlantic domain and Greenland (Eicher, 1987, Lotter et al., 1992, Ammann et al., 2013). Amsoldingersee contains the complete Late-Glacial sediment sequence, including early deglacial anoxic periods (Lotter and Boucherle, 1984).

A multi-proxy approach which includes Hyperspectral Imaging, sedimentary pigments, XRF, CNS and P, Fe, Mn and P fractions, and pollen was applied on the sedimentary record of lake Amsoldinger. Hyperspectral Imaging offers a way to reconstruct primary production, hypolimnetic anoxia and compositions of major primary producer groups at unprecedented (μm-scale) resolution on long time scales (Zander et al., 2023). Sedimentary pigments were used to investigate changes in past producer communities (Leavitt & Hodgson, 2001; Bianchi and Canuel, 2011). Pigments of anoxygenic phototrophic bacteria APBs (purple sulphur bacteria) are used as indicators for lake stratification and hypolimnetic anoxia (Züllig, 1986; Zander et al. 2022. We use XRF, CNS, and sequential extraction of sedimentary P, Mn and Fe to diagnose potential chemical feedback during events of hypolimnetic anoxia (Tu et al., 2021)."

*L120: "continuously sieved" – suggest to change to "Consecutive (or contiguous) 1-cm-thick sediment samples were sieved". As it stands you are implying that you never stopped sieving.*

➔ **Done**

*L126: Perhaps not for this paper, but why was no pollen counted for materials younger than the Allerød?*

➔ Pollen was analysed only in low resolution and in a specific core section to gain an assurance that the pollen record of AMS is similar to other published pollen records from Swiss lakes. Swiss lakes are very well-studied for their pollen

successions, and the pollen records are well-described and established. That is also why we were able to use the first occurrences of pollen as regional chrono-stratigraphical markers, specifically for the Bølling and early Allerød, when the three key tree species occur (Juniper, Beech, and Pine). Thus, as many pollen records from the Swiss plateau (Gerzensee, Moossee, Burgaschisee, and many more lakes) already exist, adding one will not change our understanding of the paleoenvironment.

*L212: Here it states RDA is carried out but it is unclear where this comes back later in the manuscript. The RoC results are reported clearly, but I didn't see the RDA explicitly reported on.*

➔ **RDA now explicitly mentioned** in the first paragraph discussion section 5.2 reading as follows:

"Redundancy analysis indicates that aquatic primary production in Amsoldingersee (e.g., Tchl, total carotenoids) covaries with TOC and follows predominantly, but not exclusively, temperature and arboreal pollen (RDA, Fig. S9)."

*L223: what does "consistent" mean in this context?*

➔ Consistent between the different dating approaches which is now added. **Reads as follows:**

"The chronology (Fig. 2) is constrained by calibrated $^{14}$C AMS ages of nine taxonomically identified terrestrial plant macrofossils (Table S1), three pollen-based biostratigraphic markers (Fig. S4), and the Laacher See Tephra (13,006±9 yrs BP, Reinig et al., 2021), yielding ages between 14.9 and 10.5 ka cal. BP consistent among the different dating approaches."

*L224: add if this is 1 or 2 sigma*

➔ **$2\sigma$, implemented**

*L271: Results 4.2: I find the description a bit confusing. I understand you go by LT rather than chronologically, but: LT4 is split up into two separate parts, with a larger unit during the BA and then a short unit at the end of the YD. LT4 (B/A) is given an interpretation, but the upper LT4 unit is not interpreted. LT5 is then split into two subunits in the "title" (L281) of 5.21-5.19 and 5.18-5.09m, but these subunits are sequential and the difference between these two parts of LT5 is never explained. I understand that in the Results section you are taking a LT by LT rather than a chronological approach, but the reader would benefit from a clearer explanation of where the LT sequence differs from the chronological sequence.*

➔ **Now clarified** in the text. Now every subsection of each lithotype has its own interpretation
➔ These difficulties arise because of the Laacher See Tephra layer which occurs with the lithotype 4-5 transition.
➔ **We have added** an introductory sentences clarifying the structure of the lithology description to the results section: "We describe the lithology in lithotypes, which are independent of depth, identified by the unconstrained clustering based on the similarities in the sediment geochemical composition. Therefore, a lithotype can be found at various depths (Fig. 3) where the sediment composition is similar and does not fully follow the chronological approach, e.g., Lithotype 4 and Lithotype 5)."

*L316: "that align with" – this seems incorrect to me. Only the Cluster 1/2 transition aligns with a lithostratigraphic transition. Cl2/3 just post-dates the lithostratigraphic transition, and Cl3/4 is smack in the middle of LT5. It's actually very interesting to see that these indicators change at different points, so I would suggest emphasising that.*

➔ **This is now explicitly mentioned** in the results section and emphasized in the discussion under the Younger Dryas header where the transition is of importance to strengthen the interpretation of the multiphasic YD.

*L350: I suggest adding "statistically insignificant" rather than just "insignificant" as you do report on and interpret these, so obviously there's some interest in this*

➔ **Done**

*L399: You do clearly show the importance of PC1, explaining a major 67% of the variability in your dataset. PC2 and PC3 are then only explaining very minor proportions at 9% and 7%, respectively. I would personally suggest rather than showing a bi-plot of PC2 vs PC3, to show one bi-plot of PC1 vs PC2, and another of PC1 vs PC3. There might be good reasons not to do this, but I think this would also visually represent the overriding importance of PC1, which is now somewhat lost.*

➔ We added a PC1 vs PC2 graph in the supplementary (this can be added as S10, as suggested by the reviewer). We have added an explanation in the text as to why PC1 is not used. As seen in the Fig S10, PC1 displays only the total productivity, e.g. it captures little of compositional changes. Because we are mostly interested in using pigment data

to asses compositional change, we are choosing to leave PC1 out of the analysis here, as total algal abundance is something different than algal composition.

➔ **Text modified to:** "In comparison, PC1 (63% of the variance, S10) follows total algal abundance (e.g. Tchl) and is, therefore, of little interest when assessing the changes in algal composition. On the other hand, when going beyond the dominant imprint of total algal abundance (PC1) we can pinpoint compositional changes."

➔ **Figure S10 added to Supplementary Material:**

➔ **PC1 vs PC3 was not added** because it would mean too much unnecessary appendices; besides if we add 1 vs 2 and 2 vs 3 then the reader has seen every pc; no need to add 1 vs 3 as well.

[Figure]

*L461: could you explain how increasing CO2 could lead to impact on the landscape or the lake in absence of changes in temperature and vegetation composition? Similarly, are there any (e.g. geomorphological) reasons for increased landscape stability in the absence of changes to the terrestrial vegetation?*

e revisited the data. If we look in detail at our pollen data (2 samples before and 2 samples after the lithological transition) we do not observe any compositional change. The *Betula* increase - that was recorded in Moossee (Rey et al., 2020) - only occurs later, at higher stratigraphical levels, excluding local vegetation-driven landscape stabilization as a cause for the transition. The uncertainty of the chironomid-inferred summer temperature reconstruction appears too large to exclude a subtle summer warming at that time.

To answer the question: It could be that the shift was caused by a local change in the geomorphology. We have several theories, but our evidence is inconclusive. Hence, these theories cannot be written in the paper. Therefore, we removed the CO2 curve from the paper.

➔ **The text now reads:**
"Amsoldingersee formed during H1, when retreating glaciers left a depression in the drumlin landscape (Fig. 1a). The sediment record starts with 2.65 m of laminated calcareous silty clay deposited into an early-deglacial oligotrophic perennial lake. Around 16.2 ka cal. BP, the sediment composition shifts to an organic gyttja. This transition marks an abrupt end of silty clay input into the Amsoldingersee basin. The sharp lithological transition can be explained as decreased sedimentation rates accompanied by increased organic matter deposition and a substantial reduction in detrital clastic input, suggesting increased landscape stability. No changes in local vegetation composition are observed and our data are insufficient to conclude on the cause of this lithological change."

*L516: Out of interest, is there a role for wind (strength) here? Is the lake currently experiencing strong winds due to e.g. tunnelling along the valley?*

➔ No, although there are weak thermal winds (drainage flow of cold air during night, only during anticyclonic synoptic situations) confined to the bottom of the Aare valley. They hardly reach up to the Amsoldinger plateau where the lake is found.

*L532-539: Personally, I find it very interesting to see the 3-stage development of the YD. As you well explain, for a long time the YD was seen as a stable event. Later, the 2-stage event interpretation got more attention following the papers by Bakke et al. (200g Nat Geosci) and Lane et al. (2013 Geology). More recently, there seem to be more studies appearing that show evidence of an even more dynamic event, including a 3-stage YD in the SE of the UK as shown by Francis et al. (in press, JQS) as well as further afield (Fastovich et al., 2022; QSR). The Weber et al. (2020) paper that you cite also shows additional complexity during the YD. Sorry this is more a note than a comment, but I do find this result really interesting!*

➔ Yes, it is interesting; I am recently part-taking in a YD working group led by Cecile Blanchet for this reason. If you are interested in joining, you can reach out to Cecile.

*L540: why is there no interpretation of the early Holocene Lake environment? Surely, it's interesting to the reader to know how the transition into the Preboreal happened, and if circumstances (e.g. Si limited) in the Preboreal were similar to those of the (late) A/B?*

➔ The focus of the paper is the Late-Glacial, with H1, the B/A warming and the cooling into the YD. The Transition to Early Holocene is beyond the scope of this paper and will be dealt with in a forthcoming paper that includes the entire Holocene and 20$^{th}$ C warming.

*L540: "what drove communities" is a bit vague – do you mean community composition, overall productivity, or both?*

➔ **Clarified and changed to: "**What drove the algal community composition?"

*Conclusions: The order of PP and anoxia is reversed in the Conclusions compared to the Discussions (where PP (section 5.2) precedes anoxia (Section 5.3); I suggest treating them in the same sequence in both the Discussion and in the Conclusion.*

➔ Changed the sequence in the conclusions.

---

## Author Comment (AC2)

Dear Dr Francus,

We would like to thank you for your constructive review which helped us to improve the clarity of our work.

Our point-by-point responses follow:

**General comments**

**Abstract:** The introduction to the topic is quite long and could be shortened.

➔ **Action:** We rephrased and shortened the introduction to the topic accordingly. **Figures and call to figures.**

In brief, please make sure that what is described in the text is easy to find in the figures.

1. While the figures are well drafted, it is often quite difficult to find the information described in the text. For instance, the text of sections 4.2, 4.3 and 4.4 refers to the chronozones H1, Bølling, Allerød, YD and Holocene, but none of the figures called for (Figs. 3-6) contains that information. I know that these figures are already very busy, but would it be possible to add these chronozones to Figs 3-6? It would make the paper easier to follow.

➔ The figures 3, 4, 5 and 6 were edited and the chronozones were added.
➔ The figures 7 and 8 were switched for better readability

2. I understand that you have established three different stratigraphies (litho-, cluster-, pigment-) and that you also use the chronozones, but the text is not easy to follow for the reader who is not familiar with the study.

➔ We clarified at the beginning of the result section how the results are structured. Similarly, at the beginning of the discussion section, we state the structure of the discussion which uses the chronozones.

"To describe our results systematically, two datasets, one high-resolution and the other low-resolution are presented using lithotypes derived from unconstrained hierarchical clustering (section 4.2) and pigment zones resulting from CONISS clustering (section 4.4), respectively."

3. Along these lines, section 4.4 is hard to follow, because it is not clear what figure the text is referring to: is it 6 or is it 7?

➔ **Action:** Adjusting the referencing of figures 6,7,8 throughout the text.

4. To make things even more complex, you also use the NGRIP stratigraphy (e.g., (GI-1d)). I understand why you do that, but is it really necessary? Since you are not really discussing the potential leads and lags of your lake with NGRIP?

➔ We agree with the reviewer's comment and removed the NGRIP stratigraphy from the manuscript.

5. Another example is at **lines 492-493**: one needs to switch from one figure to the other to follow your discussion. Here, you refer in the same sentence to PZ-2a, described in fig. 7, and to features outlined in fig. 9i.

➔ **Action:** Pigment zones were removed from the discussion text when referring to primary production or anoxia and only Figure 9 was used to support the discussion text.

**Chronology**

1. Calibrated dates. All the dates have been calibrated using Intcal20, but none of the dates mentioned in the text and the figures indicate "cal". Please add "cal" before all occurrences of BP.

   **L228:** here you should use kyr instead of ka, because it refers to some duration.

   **L618:** replace the two occurrences of "ky" by "ka", as they are dates and not durations (and add "cal" of course).

   ➔ **Action:** We corrected the age annotations thought the manuscript.

2. It is not clear how the chronozones have been defined. You write on **lines 126-127:** *"We combined 14C dates with well-dated regional biostratigraphic markers (Ammann et al., 2013; Rey et al., 2017, 2020)"*, but I'm not sure if I understand what you have done practically. Are you just relying on your age model, and using the dates of the chronozones defined by Ammann et al., 2013 (or the NGRIP chronology) to set the boundaries used here? Or do you use the changes in one of your indicators to set the boundaries of the chronozones?

   ➔ **Response**

   First, we established our chronology using 14C dates, 4 biostratigraphic marker layers (our own pollen data compared with the regional literature) and the LST.

   In the following and for better readability and orientation, we added (in figures and text) the chronozones and their ages as defined by Ammann et al. (2013).

   ➔ **Action: The text was slightly extended:** "We combined $^{14}$C dates with well-dated regional biostratigraphic markers (Ammann et al., 2013; Rey et al., 2017, 2020) by comparing pollen profiles in Amsoldingersee, with those of Gerzensee and Moossee. Previously reported ages for biostratigraphic markers (Table S1) were transferred to our core. Pollen was analysed in contiguous 2-cm intervals (Moore et al., 1991) up to the LST. The chronozones and their ages (Figs. 3, 5, 6, 8 and 9) are taken from Ammann et al. (2013).."

3. Along those lines, is your age model accurate enough to briefly discuss the timing of these chronozones? (This is probably another paper, but maybe somebody will want to do that.)

   ➔ **Response:**
   Probably not; technically, the 2σ uncertainty in our chronology is on the order of 300 years (see text). The benchmark papers in that respect are Ammann et al. 2013 and van Raden et al. 2013 who used the high-resolution δ18O record of Gerzensee to line up with the NGRIP δ18O chronostratigraphy.

**Section 4.2.**

At the end of the description of each lithofacies, there is an interpretation. I'm not against including interpretations of the lithofacies here, but one needs to make sure there is no repetition in the discussion section.

**L251:** "of low aquatic primary production (PP) and high accumulation rates of fine clastic sediments (matrix effect)." belongs to a discussion section, not a result section (see comment about section 4.2 in the general comments).

➔ We adjusted the text of section 4.2 by removing discussion-like text and by keeping only data description and data interpretation to avoid repetition in the discussion.

**Discussion**

Why is the onset of the Holocene not discussed? It was announced somehow in the introduction by saying that your study would help to understand the response of lacustrine systems in a quickly warming climate. Yet, the early Holocene is usually considered to have a very fast warming. If it is the recovery to warming that is of interest, please write it down in the text of the discussion.

➔ The study was designed to investigate one entire temperature cycle from cold (H1), to warm (B/A) and back to cold period (YD); for that purpose, we selected the Late Glacial warming and Younger dryas cooling as the target time span. Indeed, a limited fraction of our data (XRF and HIS) captures also the onset of the Holocene, but HPLC pigment data and sequential P, Mn and Fe data are missing (and were not in the scope of our study). A similar study covering the onset and entire Holocene is under way.

**Section 5.5.1.**

You are discussing AO, GZ, YD in this section dedicated to the description of H1. This is not logical. Maybe should you add a "dust record section" that covers multiple time slices.

➔ Accordingly, we moved this part to a new section 5.1.4. The Late-Glacial dust record

**Other minor comments**

**L47:** one understands you are talking about species only later. I suggest writing: "(...) promote generalist species competitiveness (...)".

➔ **Implemented**

**L105:** I think you should specify that the DOE-1 corresponds to the Bølling because I'm not sure that the readership of Biogeosciences is familiar of the DOE stratigraphy.

➔ **Thank you for your suggestion, we adjusted the text as follows:** *"Dansgaard-Oeschger Event-1 (DOE-1) started with the Bølling warming…"*

**L132-135:** I'm not sure to understand what you have done here? You XRF scanned the core three times and measured the standard deviation of your measurements? Was this performed on clr-transformed data or on raw data?

➔ **On the raw data, this is now clarified in the text:**
"For each core, triplicate scans (15 mm long) were taken at the top, middle and bottom of the core to quantify and test the standard error of elements (n = 27). Raw data of Al, Si, K, Ti, Zr, Cu, Ca, Sr, Mn, Fe, and S had a relative standard error of <15% and were considered reproducible and thus used for further analysis."

**L215:** I have not seen the R codes on Github.

**The code will be available upon paper acceptance.**

**L237:** only XRF data are shown in S5.
➔ **Correct, adjusted accordingly**

**L249:** how was water content measured? This is not in the method section.

➔ **Added:** "Water content was calculated from the sediment weights before and after freeze-drying."

**L301:** please spell out what LT means in the caption.

➔ **Figure 3 caption edited:** "Lithotype (LT)"

**L315:** not all transitions align with lithotype changes.

➔ **This is now explicitly mentioned** in the results section and emphasized in the discussion under the Younger Dryas header where the transition is of importance to strengthen the interpretation of the multiphase YD.

**L369-370:** Could the degradation of chlorophyll also be due to anoxia?

➔ To our knowledge, anoxia would rather favour pigment preservation as pigments degrade by exposure to oxygen and light. Thus, the CPI is driven by those two factors.

**L413-415:** What I understand from these lines is that the degree of degradation (PC3) is due to high grazing pressure during anoxic events. I don't understand this, as anoxia should prevent the presence of grazing organisms. Is there something I'm missing here?

➔ Grazing takes place in the epilimnion, which is always oxic; whereas anoxia refers to the hypolimnion in a well stratified and productive lake. **We clarified the text:**

**L424** "… zooplankton grazing in the epilimnion, essentially separating degradation pathways (Bianchi and Canuel, 2011; Lami et al., 2000)."

**L439:** I suggest the following: "(…) show that environmental changes on the Swiss Plateau responded very closely to large-scale climatic shifts (…)".

➔ **Implemented**

**L441:** try to be more specific about how you have done that.

➔ As is written, we contextualize our findings by discussing them along chronozones as defined by Ammann et al. (2013).

➔ **Now the text read as follows:**

"To place our findings in context with the large-scale Late-Glacial paleoclimate evolution, we discuss the development of Amsoldingersee along chronozones as defined by Ammann et al. (2013).

**L458:** I don't see anywhere a curve with sedimentation rates. This "sharp" decrease is not visible in the age model presented in Fig. 2. Actually, you have no date before 16.2 ka BP, and there is just the assumption that the lake formed around 19 ka. The sedimentation rate you infer is quite uncertain, even if it makes sense. I would be more cautious here.

➔ **We agree and adjusted the text accordingly:**

"Amsoldingersee formed during H1, when retreating glaciers left a depression in the drumlin landscape (Fig. 1a). The sediment record starts with 2.65 m of laminated calcareous silty clay deposited into an early-deglacial oligotrophic perennial lake. Around 16.2 ka cal. BP, the sediment composition shifts to an organic gyttja. This transition marks an abrupt end of silty clay input into the Amsoldingersee basin. The sharp lithological transition can be explained as decreased sedimentation rates accompanied by increased organic matter deposition and a substantial reduction in detrital clastic input, suggesting increased landscape stability. No changes in local vegetation composition are observed and our data are insufficient to conclude on the cause of this lithological change."

**L462-466:** your interpretation makes a lot of sense, but you don't have the smoking gun. A stronger argument would be if you could show that the rounded grains in your sediment are Ti, K, Si and Zr-rich and the angular ones are from carbonate rocks.

➔ Indeed, more in-depth observation, using SEM, would be required to support this argument. We analysed the grains under light and cross-polarized microscope, but the changes amongst the grains were non-quantifiable and, hence, not discussed to keep the paper somewhat short.

**L469:** How this correlation has been calculated to cope with their different resolutions? Also, it is not easy to tell if the Ti curve in Fig 9g is the low-resolution one or the opposite. More contrasted colours would help.

1. Using linear regression
2. It is the low-resolution one; the contrast is enhanced.

**L481:** I'm not sure this statement is true and useful.

➔ **Removed the statement.** "This is quite particular for a lake sediment record."

**L489:** in my view, turbid waters are not compatible with low clastic sedimentation rates, or maybe there is something I don't understand here.

➔ Not necessarily; today, in Alpine lakes during summer, clay/fine silt may mostly remain in suspension and limit Secchi depth (and color the water). Clastic varve thickness may be <<1 mm yr$^{-1}$ mostly consisting of the winter clay cap (formed during ice cover), the summer layer may be very thin. Accordingly, we replaced 'turbid water' with 'suspended solids'.

"During H1, starting at 16.2 ka cal. BP (; Fig. 7), anoxygenic phototrophic bacteria (APB) and some cyanobacteria inhabited the lake, indicating hypolimnetic anoxia during the very early phase of the lake development. APB have an advantage over other photosynthetic organisms in light-limited conditions (Karr et al., 2003); in Amsoldingersee, light could have been limited because of suspended solids in deglacial windy environments and/or prolonged winter ice cover"

**L497:** this fluctuation is barely visible. Is it a question of scale?

➔ Yes, we agree. We specified and now refer in the text to Fig. 9b, f and g.

**L514:** I do not see the 80% in this Bølling/Allerød interval. It starts near 0% and ends around 60%.

➔ In figure 9 we plotted only pollen % of individual species with the purpose of showing the vegetation changes. We added the total arboreal pollen % in the Fig. S4 and added reference to Fig. S4.

**L537:** "Drought" is quite a strong word. In this small closed lake, a drought would have a more spectacular impact. Maybe you should simply say that the climate was drier and that it produced a lowering of the lake level.

➔ **Changed sentence to:** "The end of the YD was associated with a persistently drier climate leading to lake-level lowering (Lotter & Boucherle, 1984)."

**L541:** one should expect high rates of changes between phases, but not during phases, right?

➔ Yes, we edited the text for clarity:

"Overall, algal community changes are synchronous with lithotype transitions and phases with high RoC (Fig. 7), which match the transitions of the chrono- and biostratigraphic zones reflecting large-scale climatic variations in the North Atlantic domain (Ammann et al., 2013)."

**L568:** Fig. 9g does not show Si.

➔ **Yes, it shows Ti;** We removed Si

**L569:** this idea of Si-fertilization has already been presented earlier. Try to avoid repetition

➔ **Rewriten to have less repetition:** "Further, after the substantial Si-fertilization by LST (13.05-12.90 ka BP; Fig. 7) we observe an isolated response of silicifyers, while green algae (lutein and Chl *a* - related pigments) and cyanobacteria remained invariant."

**L571:** what do you mean here? How the LST could Si-fertilize the lake before the eruption even occurred?

➔ **We edited the two sentences for better clarity:**

"Blooms of diatoxanthin (diatoms) and alloxanthin (cryptophytes) persisted during the YD, suggesting that Si fertilisation persisted throughout the YD. It is unusual to find signs of aerial silica fertilisation related to the tephra (Züllig, 1986) or dust in lake sediment records because these typically contain substantial siliciclastic fractions (Koffman et al., 2021)."

---

## Author Response (AR2)

Dear Petr Kuneš,

We adjusted the text according to your two points, the last paragraph of the intro now reads as follows:

> To answer these questions, we investigate a sediment core from a small kettle hole lake on the Swiss Plateau, Amsoldingersee, using a multiproxy approach. This lake is located adjacent to Gerzensee in an area that responded in extraordinary detail to Late-Glacial climatic changes documented in the North Atlantic domain and Greenland (Eicher, 1987, Lotter et al., 1992, Ammann et al., 2013). Amsoldingersee contains the complete Late-Glacial sediment sequence, including early deglacial anoxic periods (Lotter and Boucherle, 1984). ~~a multi-proxy dataset including Hyperspectral Imaging, sedimentary pigments, XRF, CNS, P, Fe, Mn fractions, and pollen was produced from the sedimentary record of lake Amsoldingen. Hyperspectral Imaging offers a way to reconstruct primary production, hypolimnetic anoxia and compositions of major primary producer groups at unprecedented (μm-scale) resolution on long time scales (Zander et al., 2023). Sedimentary pigments were used to investigate changes in past producer communities (Leavitt & Hodgson, 2001; Bianchi and Canuel, 2011). Pigments of anoxygenic phototrophic bacteria APBs (purple sulphur bacteria) are used as indicators for lake stratification and hypolimnetic anoxia (Züllig, 1986; Zander et al. 2022). We use XRF, CNS, and sequential extraction of sedimentary P, Mn and Fe to diagnose potential chemical feedback during events of hypolimnetic anoxia (Tu et al., 2021).~~

And regarding your second point:

> We already added in the caption of the RDA (in SOM) that the pollen data are from (Rey et al,. 2020), but now we also explicitly mention this in the methods section where RDA is mentioned.

> "For RDA (Fig. S9) we used the pollen record from Moossee (Fig 1b, Rey et al., 2020)."

Alongside, we made some minor edits in the Introduction and Method sections.

Best, Stan